RESEARCH COMMUNICATION

# Complex spike synchrony dependent modulation of rat deep cerebellar nuclear activity

Tianyu Tang[1†], Timothy A Blenkinsop[2†], Eric J Lang[1]*

[1]Department of Neuroscience and Physiology, New York University School of Medicine, New York, United States; [2]Department of Developmental and Regenerative Biology, Mount Sinai School of Medicine, New York, United States

**Abstract** The rules governing cerebellar output are not fully understood, but must involve Purkinje cell (PC) activity, as PCs are the major input to deep cerebellar nuclear (DCN) cells (which form the majority of cerebellar output). Here, the influence of PC complex spikes (CSs) was investigated by simultaneously recording DCN activity with CSs from PC arrays in anesthetized rats. Crosscorrelograms were used to identify PCs that were presynaptic to recorded DCN cells (presynaptic PCs). Such PCs were located within rostrocaudal cortical strips and displayed synchronous CS activity. CS-associated modulation of DCN activity included a short-latency post-CS inhibition and long-latency excitations before and after the CS. The amplitudes of the post-CS responses correlated with the level of synchronization among presynaptic PCs. A temporal precision of $\leq$10 ms was generally required for CSs to be maximally effective. The results suggest that CS synchrony is a key control parameter of cerebellar output.

**Editorial note:** This article has been through an editorial process in which the authors decide how to respond to the issues raised during peer review. The Reviewing Editor's assessment is that all the issues have been addressed (see decision letter).

DOI: https://doi.org/10.7554/eLife.40101.001

*For correspondence:
Eric.Lang@nyumc.org

[†]These authors contributed
equally to this work

Competing interests: The
authors declare that no
competing interests exist.

Reviewing editor: Vatsala
Thirumalai, National Centre for
Biological Sciences, India

## Introduction

The cerebellum plays a significant role in many brain functions, providing information to multiple brainstem nuclei and, via the thalamus, much of the cerebral cortex. This widespread influence makes it vital to understand the factors that shape cerebellar output, which arises primarily from the deep cerebellar nuclei (DCN). Purkinje cells (PCs) are the major synaptic input to DCN neurons, accounting for some 70–80% of the synapses onto them (*Palkovits et al., 1977*; *De Zeeuw and Berrebi, 1995*). Thus, understanding how cerebellar output is formed requires knowing the relationship between PC and DCN activity.

Although the properties of the PC-DCN synapse have been characterized in a number of studies, the rules governing the translation of PC activity into DCN firing patterns under in vivo conditions are not well understood. The GABAergic nature of PCs suggests that an inverse relationship between PC and DCN activity should exist, and indeed, manipulations that cause global changes in PC simple spike (SS) firing rates produce the predicted inverse changes in DCN activity (*MacKay and Murphy, 1973*; *MacKay and Murphy, 1974*; *Colin et al., 1980*; *Montarolo et al., 1982*; *Bardin et al., 1983*; *Benedetti et al., 1983*). However, an inverse relationship is often not seen during behavior, at least at the population level (*Thach, 1970a*; *Thach, 1970b*; *Armstrong and Edgley, 1984a*; *Armstrong and Edgley, 1984b*; *Robinson and Fuchs, 2001*; *Sarnaik and Raman, 2018*). Moreover, attempts to correlate spontaneous SS activity from individual PCs with the activity of individual DCN neurons failed to reveal any consistent correlation, despite the employment of

rigorous measures to ensure that the activity of cells located in connected regions of the cortex and nuclei was being compared (*McDevitt et al., 1987*). Thus, the translation of PC activity into DCN firing often appears not to be a simple integration of SS activity.

The complexity of the PC-DCN transform is likely due to multiple factors, such as the existence of subpopulations of DCN neurons with distinct electrophysiological properties (*Czubayko et al., 2001*; *Uusisaari et al., 2007*; *Uusisaari and Knöpfel, 2011*; *Najac and Raman, 2015*), the convergence of many PCs (and cerebellar afferents) onto each DCN neuron, and the generation of two types of action potentials, SSs and complex spikes (CSs), by PCs. In addition, the specific patterning and timing of PC activity is likely a critical factor. For example, recent work suggests that DCN firing is more effectively phase locked by synchronized than by desynchronized PC activity (*Person and Raman, 2012*; *Person and Raman, 2011*).

Here, we investigate the importance of this last factor, the timing of PC activity, and focus specifically on the role played by CS synchrony among PCs that synapse with the same DCN cell. This focus is motivated by the characteristics of the patterns of CS synchrony. Specifically, synchronous CSs occur most often among PCs within the same rostrocaudally running strip of cortex (*Sasaki et al., 1989*; *Sugihara et al., 1993*; *Lang et al., 1999*), and in particular, among PCs within the same zebrin compartment (*Sugihara et al., 2007*). This and the fact that PCs located in the same rostrocaudally running zebrin compartment project to the same DCN region (*Chung et al., 2009*; *Sugihara et al., 2009*; *Sugihara, 2011*) make it plausible that the PCs that project to the same DCN cell would also have high levels of synchronous CS activity. Moreover, we previously showed that generalized increases in CS synchrony are associated with greater inhibition of DCN activity (*Blenkinsop and Lang, 2011*). Here, we build on this earlier work by providing evidence that high levels of CS synchrony are indeed present among PCs that synapse onto the same DCN cell and that synchrony is a major parameter of their influence on the DCN cell's activity. Furthermore, we address the issue of the temporal precision needed for CSs to affect DCN activity.

## Results

Here, we investigate the relationship between CS and DCN activity in order to provide a quantitative analysis of the relationship between CS synchrony and various aspects of CS-associated modulation of DCN activity. In particular, we specifically analyze the effect of synchronous CS activity for PCs that converge onto the same DCN neuron. To do this, recordings in which at least four PCs in the electrode array could be identified as presynaptic to the DCN cell (based on analysis of their cross-correlograms) were selected from the dataset of *Blenkinsop and Lang, 2011*. The specific criteria for identifying a PC as presynaptic are given in the Methods (also see, *Blenkinsop and Lang, 2011*). Out of the original dataset (n = 717 PCs, 100 DCN cells, 35 rats), four DCN cells were found that had synaptic input from at least four crus 2a PCs in the recording array. These four DCN cells (*Figure 1A–D*) were recorded along with CS activity from 59 PCs (*Figure 1E*), of which 24 showed evidence of a synaptic connection to the recorded DCN cell (henceforth referred to as presynaptic PCs), and 35 did not (non-synaptic PCs) (n = 3 rats). Of the four DCN cells investigated in the present report, the locations of two were localized histologically to the interpositus nucleus. The other two were not histologically localized but were likely in either the interpositus or medial dentate nuclei, given the distribution of recordings in the prior study. All four DCN cells had spike trains that consisted of a mix of tonic and bursting firing patterns (*Figure 1F*).

To test whether being identified in the presynaptic group reflected a difference in CS firing rates, the average firing rates of the presynaptic and non-synaptic PCs were compared. No significant difference was found (presynaptic, 0.96 ± 0.78 Hz, n = 24; non-synaptic, 0.94 ± 0.71 Hz, n = 54; p=0.91); note that 19 PCs were recorded with two DCN cells and their firing rates in each recording session are included in the totals; however, including the firing rates of those PCs from only one of the sessions did not result in significant differences between the groups (p=0.41, n = 17 and 41, and p=0.44, n = 18 and 40). The firing rates of the four DCN cells on average (34.19 ± 7.01 Hz) were lower those of our larger sample that was reported previously (45.3 ± 21.4 Hz; n = 100; p=0.037; *Blenkinsop and Lang, 2011*); however, they were all still well within the range of rates typical for DCN cells. In sum, the basic firing rates of the PCs and DCN cells identified as being synaptically-connected were typical of these cell types.

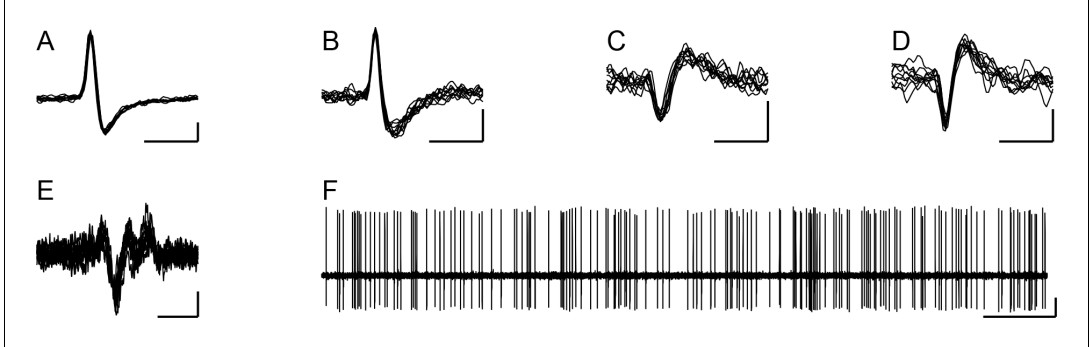

**Figure 1.** Extracellular records of DCN neurons. (**A–D**) Extracellular records (10 overlapped sweeps) showing the spike waveforms of the DCN neurons whose activity is analyzed throughout the paper. The letter of each panel corresponds to the experiment name used throughout the remainder of the paper for analyses related to that cell. Horizontal calibration bars are 1 ms. Vertical calibration bars: A, 200 μV; B-D, 100 μV. (**E**) Example of CS activity recorded from one of the PCs identified as presynaptic to a DCN cell (10 overlapped spikes). Calibration bars are 5 ms and 50 μV. (**F**) Example recording of DCN neuron (same as in panel A) to show typical firing pattern consisting of both tonic and bursting activity. Calibration bars are 0.5 s and 200 μV.

DOI: https://doi.org/10.7554/eLife.40101.002

Examples of CS-triggered correlograms illustrating the relationship between CS activity in presynaptic PCs and DCN activity are shown in *Figure 2*. In every case, a sudden sharp drop in activity occurs just after the CS (dashed lines at a latency of 0 ms), which was the criterion for identifying a monosynaptic connection between the PC and DCN neuron (the precise timing of the inhibition onset is seen most clearly in the lower row of histograms, which show the central portion of the histograms in the top row with an expanded time scale). In a few cases (n = 4 out of 24), a brief excitation preceding the CS (*Figure 2B*) was present and likely reflected the excitation of the DCN cell by collaterals of the same olivary axon that was evoking the CSs (*Blenkinsop and Lang, 2011*). However, because this response was present so rarely, we did not analyze it further. In addition to these short-latency effects, the histograms often also showed longer latency changes in the DCN activity associated with CSs. In particular, there were periods of increased DCN activity roughly 200–500 ms

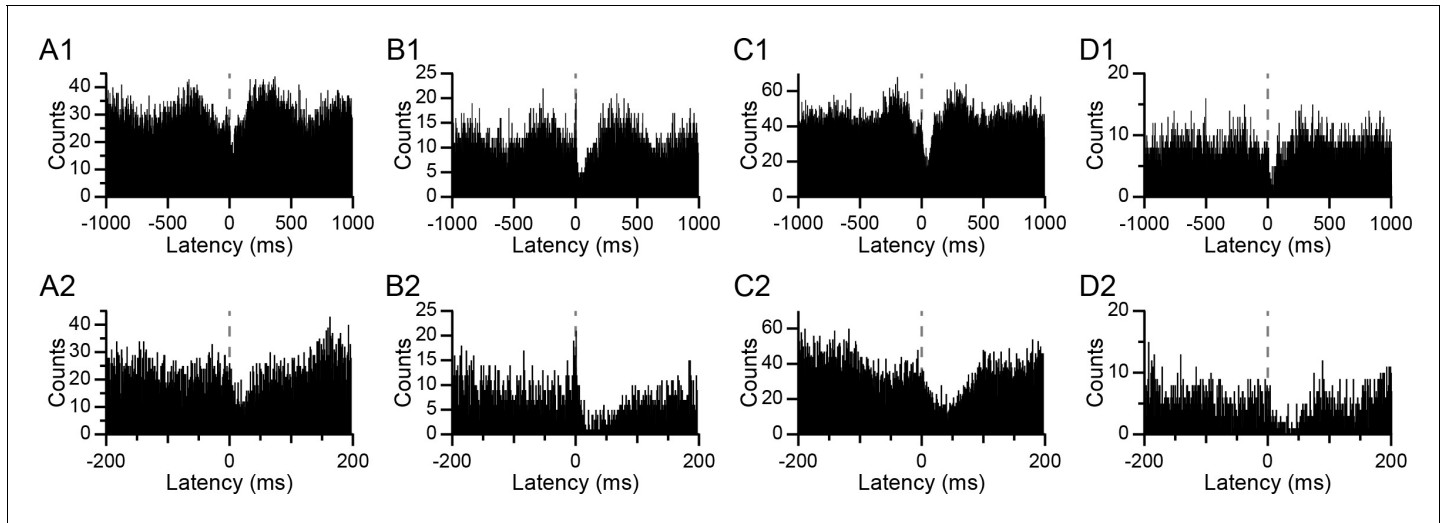

**Figure 2.** CS-associated changes in DCN activity. (**A–D**) CS-triggered histograms of DCN activity for four different PC-DCN pairs identified as synaptically-connected. Panels A and B show histograms from the same DCN cell with two different PCs. Panels C and D show histograms from two other DCN cells. The bottom row (**A2–D2**) shows the corresponding histograms of the top row (**A1–D1**) with an expanded time scale in order to see the timing of the onset of the inhibition in DCN activity at the time of the CS (latency = 0 ms; dashed line). Bin width = 1 ms.

DOI: https://doi.org/10.7554/eLife.40101.003

before and after a CS. The amplitude of both the short- and long-latency modulations varied between experiments and between PCs within an experiment, as can be seen by comparing the correlograms in *Figure 2*. Characterizing how these modulations vary with CS synchrony is the primary focus of the remainder of this paper; however, we first describe the patterns of synchrony among presynaptic PCs, because they provide further evidence in support of the correlogram-based identification of presynaptic PCs and because these patterns define the range of synchronous activity generated by the olivocerebellar system.

## PCs that project to the same DCN neuron form narrow rostrocaudal bands

PCs connecting to the same DCN cell would be expected to be clustered in the mediolateral direction (i.e., along the longitudinal folial axis), because of the topography of the PC-DCN projection (*Chung et al., 2009*; *Sugihara et al., 2009*; *Sugihara, 2011*; *Voogd and Bigaré, 1980*). The actual spatial distributions of the presynaptic PCs for each of the four DCN neurons are shown in *Figure 3A–D*. (Note that for the remainder of the paper each experiment will be referred to by the letter of the panel that shows its PC array in *Figure 3*, and that the panels in *Figure 1* showing the DCN spike waveforms also correspond to this naming convention.) In each panel, the overall

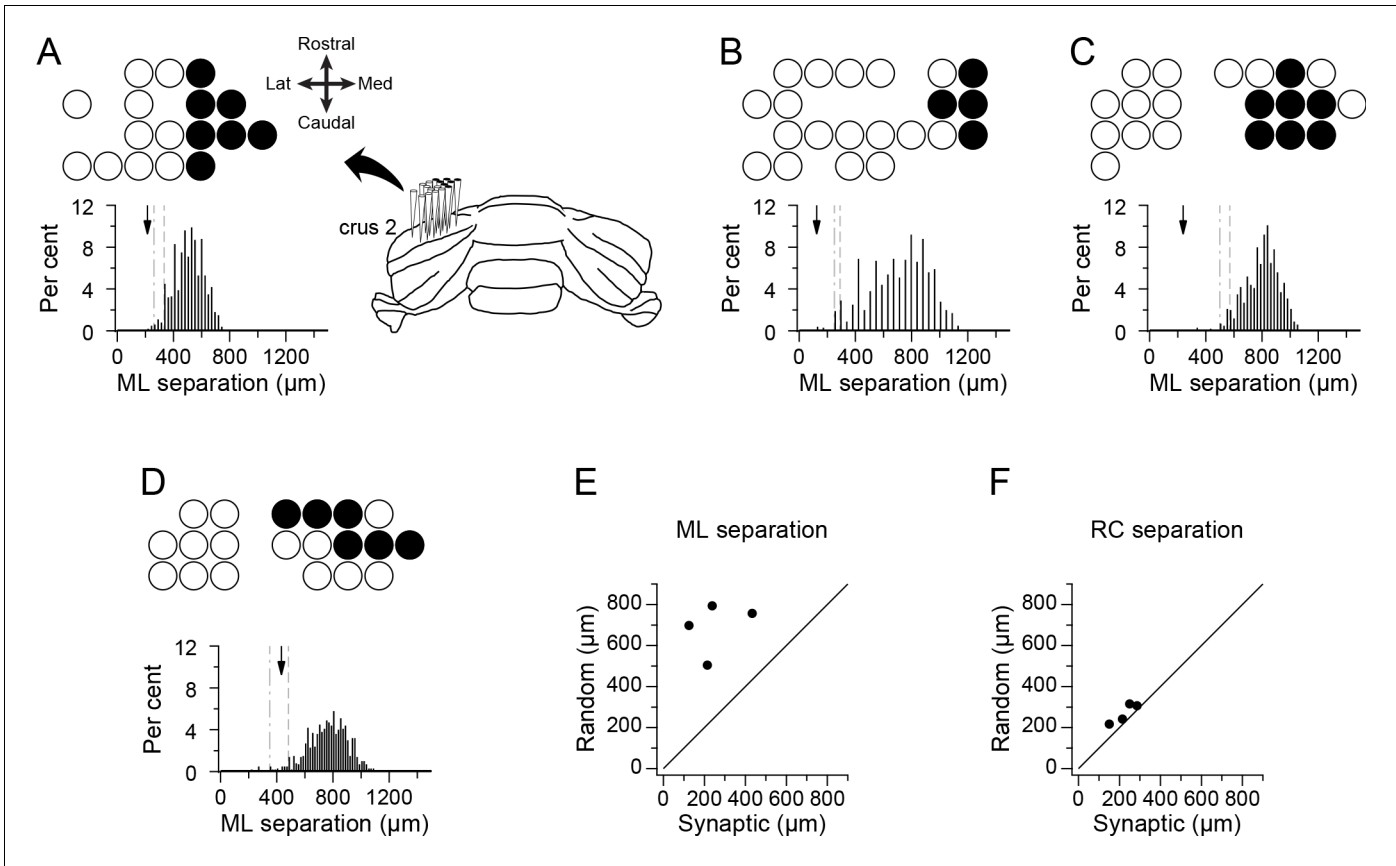

**Figure 3.** PCs identified by correlogram analysis as projecting to the same DCN neuron form spatially restricted groups. (A–D) Schematics showing recording arrays from four experiments. Circles indicate positions of recorded PCs; filled circles indicate positions of PCs identified as presynaptic to the recorded DCN cell. The histogram below each schematic shows the distribution of the mean mediolateral separation among cells from groups having the same number of cells as the presynaptic group but whose locations on the array were randomly chosen. Arrow indicates the mean separation value for presynaptic group in each experiment. The dashed and dash-dotted lines indicate the 5% and 1% percentiles, respectively. (E) Scatter plot of the mean mediolateral separation of PCs in the presynaptic group versus the mean of the distribution for the random groups. (F) Same as (E) except that the separation in the rostrocaudal direction is plotted. Note that each experiment shown in panels A-D will be referred to throughout the paper and in other figures with reference to their panel designation in this figure.

DOI: https://doi.org/10.7554/eLife.40101.004

recording array on crus 2a is indicated by the distribution of circles in each panel, where the filled circles represent the presynaptic PCs. The orientation of the recording arrays on crus two is shown by the schematic of the cerebellum in panel 3A. Visual inspection of the circle plots suggests that the presynaptic PCs are, in fact, restricted in the mediolateral direction. To test this statistically, we compared the average separation among PCs in the synaptic groups to that in groups that had an identical number of cells but in which cell locations were randomly chosen throughout the array (only one cell per location). One thousand such random groups were generated for each experiment, and the resulting distributions of average mediolateral separation between cells are shown by the histograms in *Figure 3A–D*. The dashed and dash-dotted lines indicate the 5th and 1st percentiles of the random group distributions, respectively. The arrow indicates the average separation of PCs in the presynaptic group, which falls below the 5th percentile in all cases and below the 1st in three out of four cases. Further evidence of the spatial restriction in the mediolateral direction is shown by the scatterplot of the average separation distances for the presynaptic and random groups for the four experiments, where all of the points fall above the x = y line (*Figure 3E*). In contrast, the average separation along the rostrocaudal axis was similar for the presynaptic and random groups (*Figure 3F*). In sum, the spatial extent of each presynaptic PC group is consistent with all of the PCs in the group projecting to the same DCN neuron, supporting their identification as presynaptic by the cross-correlogram analysis.

## PCs that project to the same DCN neuron show high levels of CS synchrony

In light of the spatial organization of the presynaptic PC groups, and as previous work shows that synchronous CS activity occurs among PCs aligned in the same rostrocaudally running strip of cortex (*Bell and Kawasaki, 1972*; *Sasaki et al., 1989*; *Lang et al., 1999*), we investigated the patterns of synchrony for the presynaptic groups. The level of synchrony and the mediolateral separation distance were computed for each PC pair in the arrays from all four experiments. Average (mean) synchrony was then plotted as a function of separation distance for the presynaptic and non-synaptic PCs (*Figure 4A*; presynaptic group PCs: for 0, 250, 500, 750, and 1,000 μm, n = 16, 31, 13, 2, and 1; nonsynaptic group PCs: for 0, 250, 500, 750, 1,000, 1,250, 1,750, and 2,000 μm, n = 43, 95, 69, 51, 32, 29, 10, and 3). Comparison of the mean synchrony curves for the two cell groups shows that much higher synchrony was present at all separation distances for presynaptic PCs (0 μm, p=0.04, n = 16 and 43; 250 μm, p=0.0005, n = 31 and 95; 500 μm, p=0.02, n = 13 and 69). Tests for normality (Kolmogorov-Smirnov) showed that the distributions were normal at all separation distances for

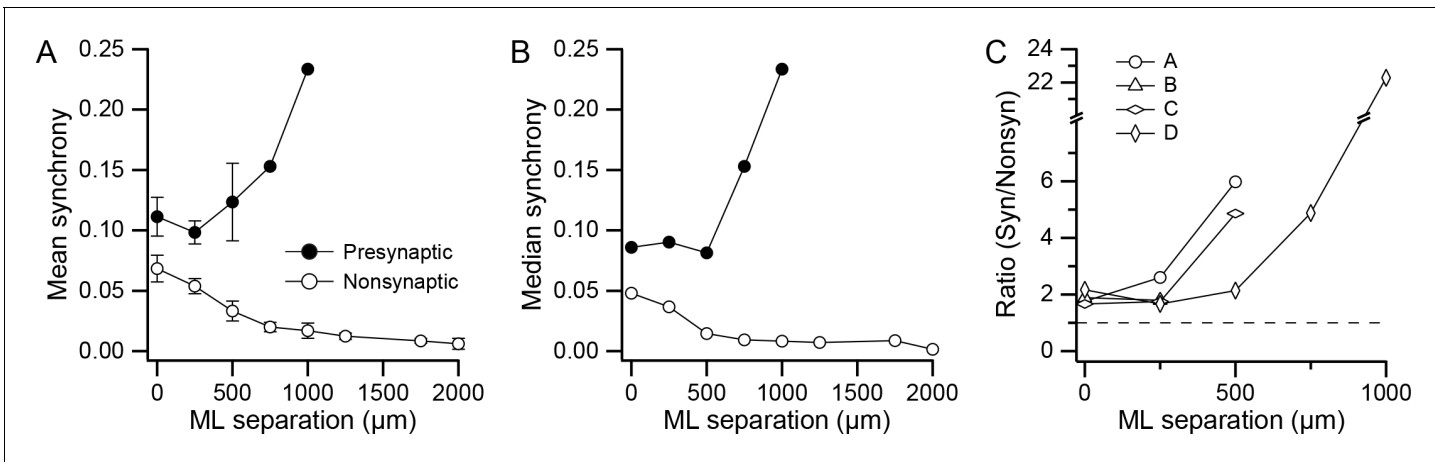

**Figure 4.** DCN projection groups show high CS synchrony levels. (A) Plot of mean synchrony as a function of the mediolateral (ML) separation between PCs from all experiments. The filled circles show the mean synchrony for pairs in which both PCs were in one of the identified presynaptic groups and the unfilled circles show the synchrony for pairs from the remaining cells in the arrays. Error bars are ±1 SEM. (B) Similar to A, except that median synchrony is plotted as a function of ML separation. (C) The ratio of mean synchrony of the synaptic and non-synaptic PCs is plotted as a function of ML separation for each of the four experiments.
DOI: https://doi.org/10.7554/eLife.40101.005

the presynaptic PC pair populations but not for the nonsynaptic PC pair populations. Thus, median synchrony values were also plotted, and they showed an essentially identical result. In addition, a non-parametric test of the presynaptic and nonsynaptic synchrony distributions showed that they were significantly different (0 µm: p=0.002, n = 16 and 43; 250 µm: p=3×10$^{-07}$, n = 31 and 95; 500 µm: p=2×10$^{-07}$, n = 13 and 69; Wilcoxon-Mann-Whitney two sample rank test).

The consistency of higher synchrony among PCs in the presynaptic group across experiments is shown by curves in *Figure 4C*, which plot the ratio of the mean synchrony curves for the presynaptic and nonsynaptic PC cell pairs. In all cases where it is defined (n = 13; 4 at 0 µm, 4 at 250 µm, 3 at 500 µm, 1 at 750 µm, 1 at 1,000 µm), the ratio is above one.

In *Figure 4*, it can also be seen that for the nonsynaptic PCs, synchrony falls off with increasing ML separation, consistent with previous studies (*Sasaki et al., 1989*; *Lang et al., 1999*). In contrast, for the presynaptic PCs, synchrony does not decline, but rather stays relatively constant (0–500 µm) or rises (750–1,000 µm) with increasing ML separation, though the n is small for these larger distances. This divergence in the two synchrony curves was present in the experiments where there were separations of at least 500 µm between synaptic PCs (Exps. A, C, and D), which is demonstrated by the increase in the synchrony ratio with separation for these experiments (*Figure 4C*). In sum, synchrony is better maintained among PCs within the same presynaptically-connected group.

The above analyses considered all of the PCs pairwise to derive the average or median levels of synchrony between PC pairs. We next investigated the prevalence of more widespread synchronization among presynaptic PCs. For this analysis, each PC was considered as a reference in turn. Each CS fired by the reference cell was assigned a synchrony level based on the number of PCs in the presynaptic group firing CSs within a specific time window (±5 ms) centered on the time of the CS in the reference cell. The distributions of synchrony levels for each PC and the average of all PCs in the group are shown for each experiment in *Figure 5A–D*. The average curves for all of the experiments are replotted together in *Figure 5E* for direct comparison. In general, events comprising CSs in only one or a minority of the PCs in the presynaptic group were most common; however, synchrony events at all levels occurred in the experiments.

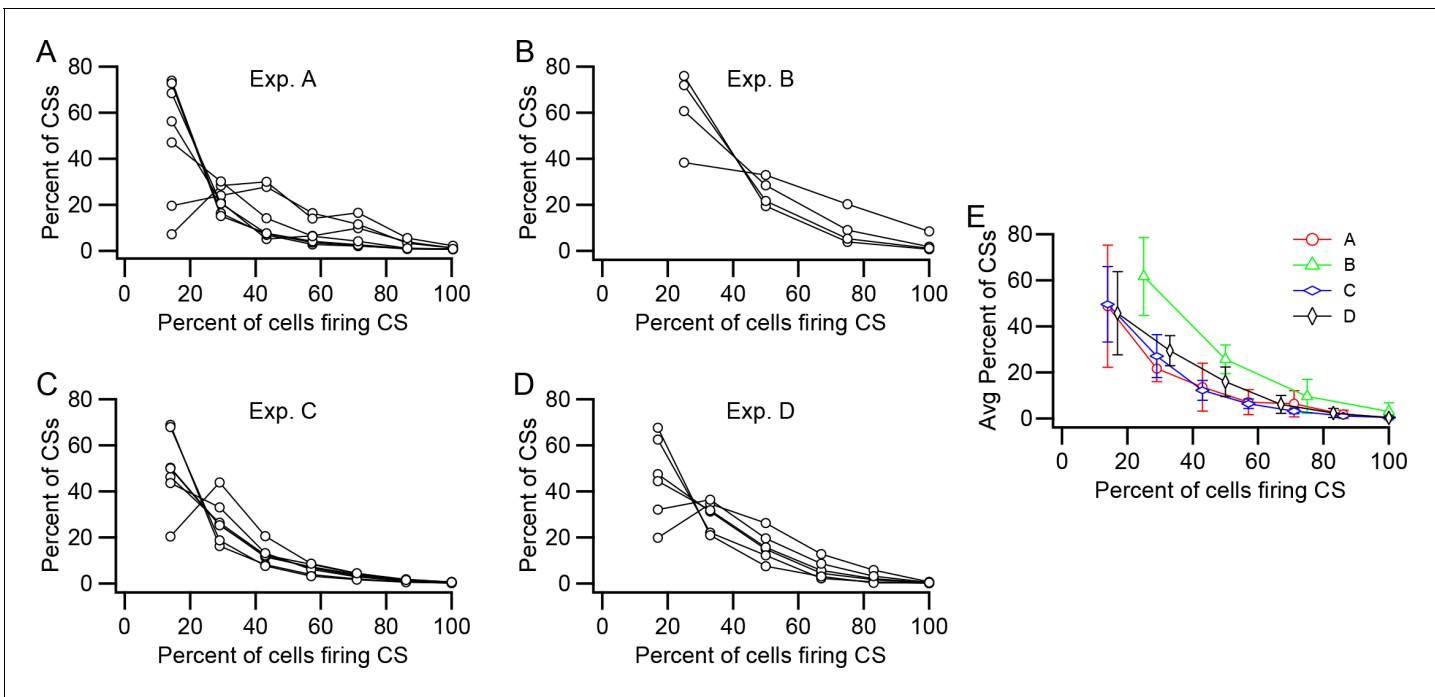

**Figure 5.** Relative prevalence of synchronous CS activity. (**A–D**) Plots of the percent of CSs at each synchrony level (Percent of cells firing CSs) within the presynaptic group. Each panel shows data from one experiment. Each curve shows the distribution of CSs for one PC in the presynaptic group. The number of PCs in each group is as follows: A, 7; B, 4; C, 7; D, 6. (**E**) Average distribution for each experiment.
DOI: https://doi.org/10.7554/eLife.40101.006

In sum, the wider distribution and generally higher levels of synchrony, and in particular, the presence of synchrony events involving most or all cells, that was found for the presynaptic PC groups indicate that individual DCN neurons are often subjected to synchronized CS input, raising the question of what is the effect of such activity on DCN cells.

## Inhibition of DCN activity varies with level of CS synchrony

CS-triggered rasters were used to begin investigating the relationship between CS synchrony and DCN activity (*Figure 6A*). Visual inspection of the rasters revealed the presence of CS-related modulation of DCN activity at a variety of time scales. To investigate the relationship of these modulations

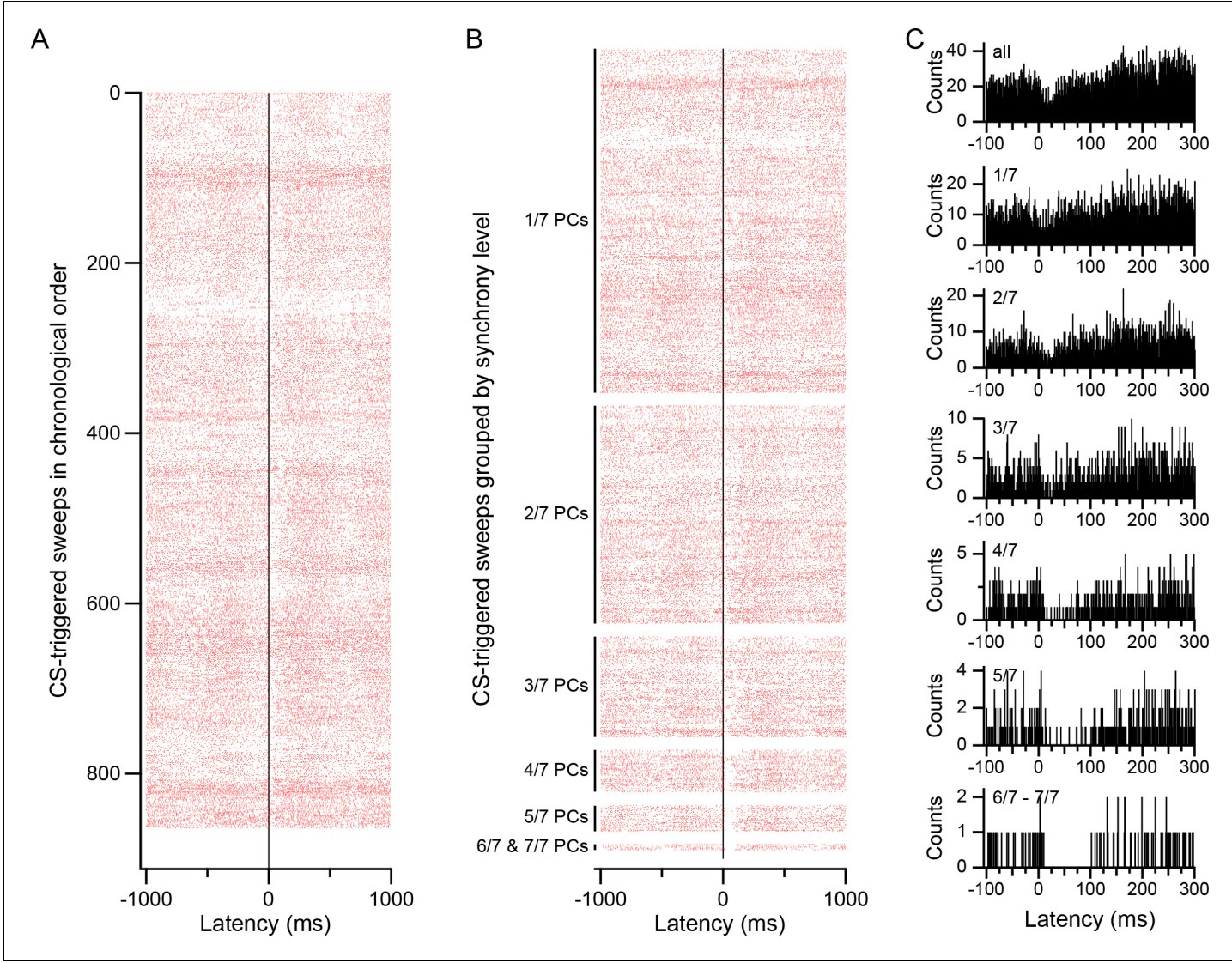

**Figure 6.** CS-associated modulation of DCN activity. (**A**) CS-triggered raster display of DCN activity. The raster was generated from all CSs of one PC during a 20 min recording (firing rates: CS, 0.70 Hz; DCN cell, 30 Hz). Each row represents the activity surrounding one CS. The rows are arranged in chronological order (top to bottom). (**B**) Raster of same data as in A, but CSs are first grouped according to synchrony level, as indicated by labels on y-axis. Within each synchrony level the rows are arranged chronologically. In particular, note how the inhibition of DCN activity just after the CS (latency = 0 ms) increases with synchrony. (**C**) CS-triggered histograms of DCN activity for different synchrony levels. Top histogram (all) generated from all CSs and corresponds to the entire raster in A and B. The remaining histograms were generated from their corresponding groups in the raster of panel B. Note the expanded time scale compared to the rasters in order to highlight the change in the inhibition that follows the CS, and the transient excitation just preceding the CS (visible in bottom four histograms).
DOI: https://doi.org/10.7554/eLife.40101.007

to CS synchrony, CSs were sorted according to the fraction of PCs in the presynaptic group that were firing simultaneously (synchrony level), as shown in *Figure 6B*. Comparison of the rasters across synchrony levels showed the presence of a strong relationship between synchrony and the short-latency inhibition that follows a CS. That is, when a PC fired a CS and the other PCs in the group did not, the post-CS inhibition was very weak, being barely visible in the raster display (*Figure 6B*, 1/7 PCs), and the histogram compiled from these CSs showed only a shallow depression following the CS (*Figure 6C*, 1/7). However, the inhibition of DCN activity in this period increased significantly with synchrony, which can be seen by comparing either the rasters or the histograms across synchrony levels. Indeed, at the highest synchrony levels (6/7 and 7/7), there is almost a complete silencing of the DCN cell for ~100 ms following a CS.

CS-triggered histograms of DCN activity were used to quantify the relationship between CS synchrony and the short-latency inhibition seen in the raster displays. For each synchrony level, a histogram was made using the CSs of each presynaptic PC, as shown in *Figure 6C*. The individual cell histograms for each synchrony level were then averaged to get the overall group response at each level. In this analysis and in succeeding ones, a synchrony level generally corresponds to a specific fraction of PCs in the group firing a CS; however, the number of events at the highest synchrony levels was small for experiments A, C and D (*Figure 5* and *Figure 6B,C*, levels 6 and 7), and so for these experiments, the highest 2–3 synchrony levels were collapsed into one 'high' synchrony level, such that every PC in the group had at least 10 high synchrony events (Exp. A: levels 5–7; Exp. C: levels 6–7; Exp. D: levels 4–6). The value assigned to this high synchrony level was a weighted average of the combined synchrony levels (the weights corresponded to the total number of CSs at each level summed across all PCs in the group).

The inhibitory effect of the CSs was then measured as the percent change in the average height in the histogram during the response period (defined below) from baseline. The baseline was defined as the average height of the histogram at latencies of $-55$ to $-5$ ms, except for experiment D, where the latencies $-30$ to $-5$ ms were used (because the correlograms were relatively flat over a shorter period preceding the CS in this experiment).

The response period limits were determined in two ways. In the first, the response period was determined using the CS-DCN histogram constructed from all CSs in all PCs of the group. The start of the response was defined as the time after the CS (t = 0 ms) when the histogram first fell below the baseline level (a moving average of three 1 ms bins was used, with the central bin being the time point). The end of the response was defined as the first point after the start time that the moving average crossed back above baseline. Thus, the duration and timing of the response period were held constant across synchrony levels, allowing the same time period to be compared. In all experiments, the start of the response, thus defined, occurred at a latency of 1.5 ms from the CS, consistent with a monosynaptic connection from the PCs to the DCN cell. The end of the response occurred at a latency of $68.75 \pm 12.66$ ms (n = 4; *Figure 7A*, All), giving an average response duration of 67.25 ms.

In the second method, the response period was determined individually for each synchrony level. The same procedure as in the first method was followed, except that the start and end of the response were defined using the individual synchrony level histograms instead of the all-CSs histogram (three-point averages were again used, except in experiment C, where the sparseness of the higher synchrony levels required a moving average of six points to obtain limits that reasonably agreed with those based on visual inspection). Although less reliable than the first method, the second allowed us to test whether the response duration varied with synchrony.

For most synchrony levels, the second method produced similar estimates of the response duration to that obtained by the first method (*Figure 7A*, compare the All value with those at different synchrony levels for each experiment). Indeed, in two experiments, there was no significant correlation between synchrony level and response duration. In the other two experiments (experiments A and C), the correlation was significant (p<0.05, n = 7 in both cases); however, most of the variation resulted from a large jump in duration at the highest synchrony levels for which the values are the least reliable, because the histograms corresponding to these levels are constructed from relatively few sweeps (e.g., compare the number of sweeps per level in the raster of *Figure 6B*). In sum, the duration of the initial inhibition of DCN activity following a CS generally does not appear to vary significantly with synchrony, except possibly at the highest levels.

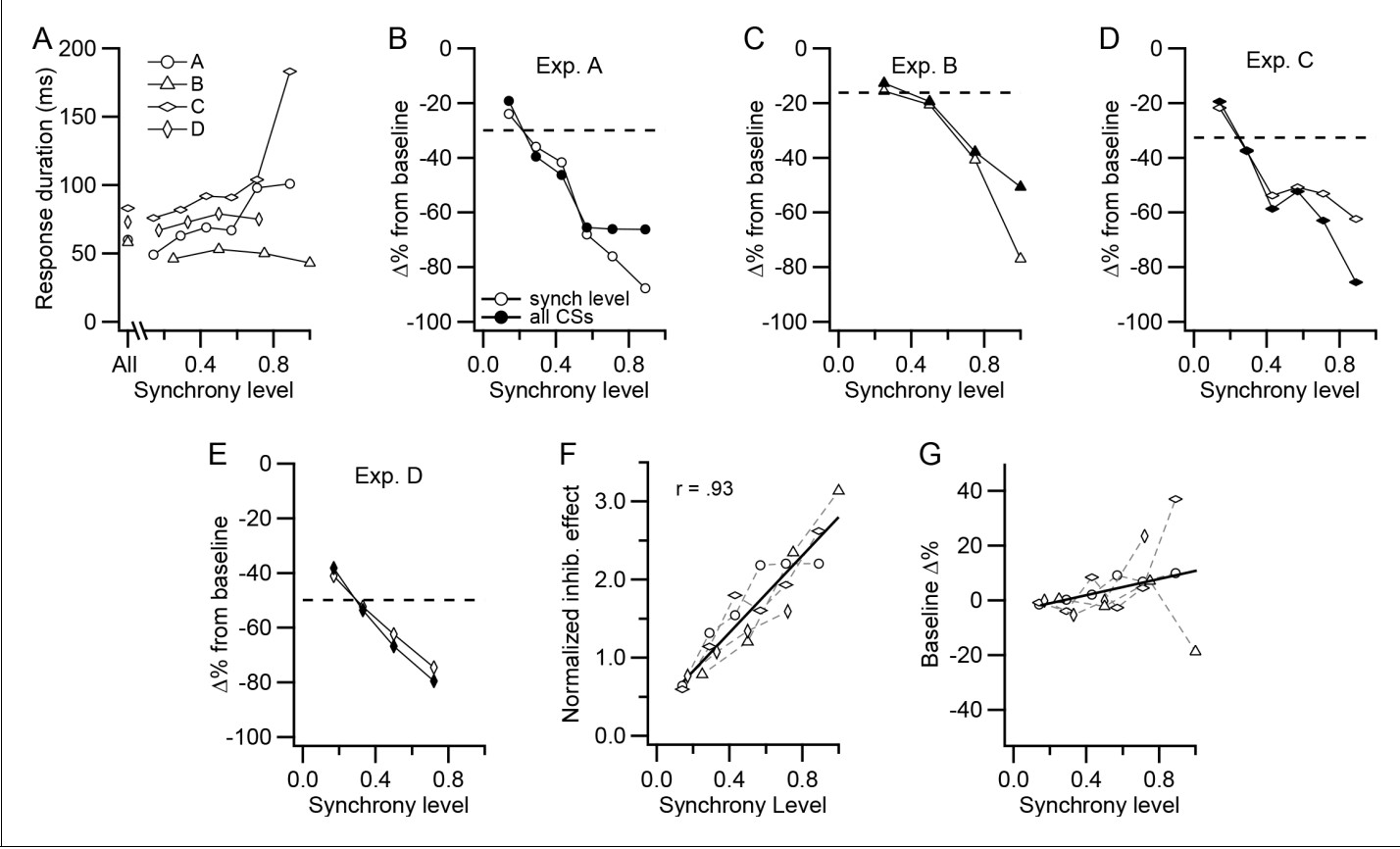

**Figure 7.** CS-associated inhibition of DCN activity is a function of synchrony. (A) Duration of post-CS inhibition is plotted as a function of synchrony level for each experiment. The durations were determined from histograms generated from the CS activity of all PCs in a presynaptic group. The points plotted above 'All' are durations determined from the histogram generated from all CSs. The other points were generated from all CSs at a particular synchrony level. (B–E) For each experiment, the percent change in DCN activity during the post-CS inhibition period from baseline activity is plotted as a function of synchrony level. The time of the response period was set either based on the All-CS histogram (filled circles) or by measurement of the duration on the specific synchrony level histograms (unfilled circles). Dashed lines indicate the change in activity measured from the All-CS histogram. (F) Comparison of the effect of synchrony across all experiments. The 'all CSs' curves in panels B-E were normalized by dividing each by the corresponding average synchrony for that experiment (horizontal dashed line in each panel). Symbols for experiments are the same as in panel A. Solid line is least squares regression line. (G) Change in the level of the baseline activity with synchrony. Histograms were normalized for number of CSs, and then baseline activity at each synchrony level was compared to the baseline level for the All-CS histogram. In some of the experiments, the highest synchrony levels were combined because there were too few events (<10) at the highest levels to analyze. In these cases, the synchrony level was set to a weighted average of the combined levels: experiments A and C, levels 6-7/7; experiment D, levels 4-6/6. No levels were combined for experiment B. Solid line is least squares regression line.

DOI: https://doi.org/10.7554/eLife.40101.008

We next quantified the change in the amplitude of the short-latency inhibition with synchrony. With either method for defining the response period, the inhibition of DCN activity relative to the baseline became stronger with increasing synchrony (*Figure 7B–E*). In all experiments, the inhibitory effect was weakest for events in which only one PC of the presynaptic PC group fired a CS and strongest at the highest synchrony level. Because no consistent variation of response duration with synchrony was found, the response duration defined by the first method is used hereafter, because of its greater reliability.

Although the amplitude of the short-latency inhibition varied with synchrony in every experiment, the average inhibition (i.e., that produced by all CSs and indicated by the dashed lines in panels 7B-E) varied between experiments. So, to compare the synchrony dependence across experiments, the inhibitory effect curve from each experiment was normalized by dividing it by the average inhibition in the experiment (*Figure 7F*). The combined data from the four experiments show a strong correlation between CS synchrony and the strength of the inhibition of DCN activity for the entire dataset

(r = 0.93, p = $4\times10^{-9}$, n = 20). Indeed, that the data points from all experiments are fit well by a single regression line suggests that the synchrony dependence was similar for all of the DCN cells recorded, despite the variations in the absolute strength of the inhibition.

Because the inhibitory effect was measured with respect to the baseline DCN activity that preceded the CS, it is possible that it could in part reflect a variation of baseline activity with synchrony. To test this possibility, the baseline activity in all histograms was normalized for (divided by) the number of CSs used to generate them and then compared (*Figure 7G*). In general, there was little change in the baseline across synchrony levels. Only at the highest synchrony levels did relatively large changes sometimes occur, but there was no clear directionality to these changes across experiments, and the slope of regression line was not significantly different from zero (p = 0.14, n = 20). Moreover, even if the slope had been significant, it would have accounted for only a 12.4% change in DCN firing rate across the synchrony levels, in contrast to the ~80% or greater decreases that were generally found.

In sum, the results suggest CS triggered inhibition of DCN activity is strongly dependent on the level of synchronization among the PCs projecting to a DCN cell.

## Synchrony determines the inhibitory efficacy of almost all PCs

We next analyzed how the dependence of the short-latency inhibition on CS synchrony varied between individual PCs. For each PC, the inhibitory effect on DCN activity based on all of its CSs was compared to that of its high synchrony CSs as defined earlier (*Figure 8*). The inhibition caused by the high synchrony CSs was significantly greater than that caused by all of the CSs that a PC fired (black data points) (All: −38.4 ± 15.4%; Synchronous: −72.5 ± 13.1; p =$1\times10^{-7}$, n = 24, Wilcoxon signed rank test). Moreover, for all PCs but one (n = 23/24), highly synchronized CSs caused greater inhibition of DCN activity. In the remaining case, the difference was negligible. Thus, the synchrony dependence for inhibiting DCN activity holds not only on average, but essentially for all PCs.

## What is the effect of non-synchronized CS activity?

For the group analysis, the lowest synchrony level showed the presence of a substantial inhibition (*Figure 7*). This was also true for the individual cell analysis (data not shown). This is most likely due to the fact that one PC represented a significant fraction of the presynaptic group (0.14–0.25), and its firing a CS thus likely corresponds, on average, to the synchronous firing of a similarly significant fraction of all of the PCs that project to a DCN cell.

But does a single CS (i.e., non-synchronous CS activity) have a significant effect on DCN activity? To address this question, we estimated the inhibitory effect of an individual PC firing a CS by fitting regression lines to the data shown in *Figure 7B–E* (lines were fit to the solid symbols). The regression analysis produced good fits in all cases ($r^2$: 0.85, 0.97, 0.91, and 0.99 for Exps. A-D). Next, the fraction of the population represented by a single PC was estimated to be

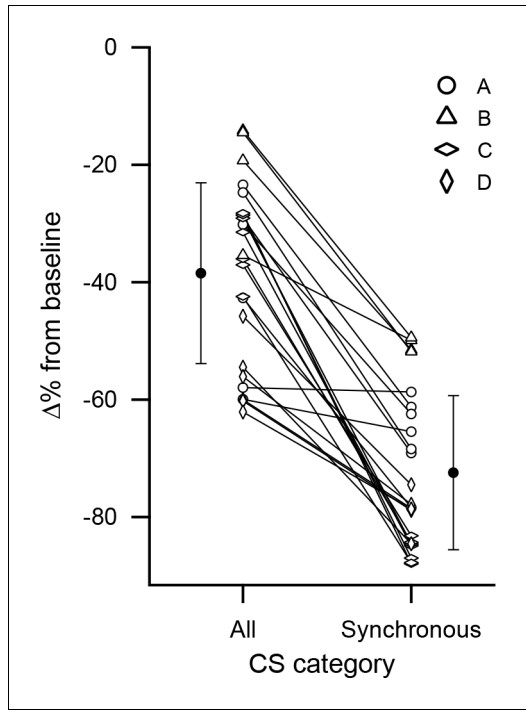

**Figure 8.** Cell by cell comparison of the inhibitory effect on DCN activity of synchronous CS activity. The percent change from baseline DCN activity following a CS was calculated for each PC based on all of its CSs or only those that were highly synchronous. The percent change values for each cell are connected by a line. The black circles indicate the averages of the two conditions. Error bars are SD. The highest synchrony levels were combined to ensure all cells in an experiment had at least 10 events: experiment A, levels 5-7/7; experiment C, levels 6-7/7; experiment D, levels 4-6/6. No levels were combined for experiment B.
DOI: https://doi.org/10.7554/eLife.40101.009

0.02–0.01 under the assumption that 50–100 PCs converge onto a DCN cell. Using these values, the regression equations yielded almost identical estimates of the inhibitory effect, which were not significantly different from zero (0.02: −15.3 ± 12.9%, p=0.10; 0.01: −14.6 ± 12.8%, p=0.11; n = 4 experiments). Moreover, for each experiment individually, the estimate of the inhibition from a single PC firing a CS was not significantly different from zero in three of four cases (p>0.05, n = 6, 6, 4; Exp. D was the exception, p=0.01, n = 4 synchrony levels). Thus, the action of an isolated CS appears to have at most a weak effect on DCN activity.

## Effect of varying the definition of CS synchrony

The prior analyses used a ± 5 ms window to define synchronous CSs relative to the CSs in a reference PC. This duration was chosen because it is on the same order of precision as indicated by the width of the central peak in correlograms published in prior multielectrode studies of CS activity (*Sasaki et al., 1989*; *Lang et al., 1996*). Nevertheless, from the perspective of the DCN cell, a different definition (or definitions) of synchrony might be warranted. Thus, we investigated the dependence of the relationship between synchrony and the short-latency inhibition on the time window used to define synchrony.

We first assessed the number of high-synchrony events in differently sized time windows (*Figure 9A*, histograms). The histograms plot the number of events in a given window that did not meet the criteria to be in smaller (or larger) windows (window size was increased in steps of ±5 ms; thus, each histogram bin has a total duration of 10 ms). In all experiments, the initial bin, corresponding to the ±5 ms window, was the highest and was followed by a sharp drop in bin height. Each experiment's histogram also showed a secondary peak, which was usually small (experiment C being the exception) at windows of ~80–120 ms, which might reflect the ~10 Hz rhythmicity sometimes displayed by CS activity. Of note is that windows between 10 and 80 ms had fewer than the average number of events. The relative rarity of events for these time windows is also observable in shallower slopes of the cumulative summation (cusum) curves for these windows, particularly for experiment D (*Figure 9A*, gray trace). The value of the cusum curve at each time window is the sum of all events that occur in that time window plus each of the smaller time windows (i.e., the sum of the histogram bars preceding and including the one corresponding to the value of cusum time window).

The prevalence of the precise synchrony events (i.e., those with the ±5 ms window) and the relative rarity of events in the next smallest set of windows, suggests that these precisely synchronized CSs are a distinct pattern of activity that could have particular significance with regard to DCN activity. To test this possibility, we compared the inhibitory effect of the high synchrony events (i.e., those events in which all or almost all PCs in the presynaptic group fired CSs, as defined earlier for each experiment) when defined using different time windows (*Figure 9B*). Note that each time window includes all high synchrony CS events given the limits of that window, including those that meet the criteria for smaller windows. For large time windows, the curves in *Figure 9B* approximate the average inhibitory effect calculated for the entire population of CSs (dashed lines), as expected, because most CSs will meet the criteria to be included in these windows. Of more interest is the window at which the curves begin to rise, as this indicates when inhibitory effect due to synchronization of CS activity begins to dissipate. In three of the experiments (B, C, and D), the curves show an immediate or near immediate sharp upward deflection, starting to rise either between the ±5 and ±10 ms windows (exps. C and D) or between the ±10 and ±15 ms windows (Exp. B). This upward deflection is followed by a continued general rise that can be interrupted by regions where the slope of the curve is reduced or the curve even plateaus (this is easiest to see for exps. B and D). The plateaus occur because essentially no new synchrony events were added for those time windows, as is shown by the shallow to flat slopes of the cusum curves over the range of those windows (*Figure 9A*, gray traces). Experiment A shows a somewhat distinct pattern in that its curve doesn't start to rise consistently until the ±45 ms window (a possible explanation for this delayed rise is given in the Discussion). Nevertheless, once it does start rising, it shows the same general pattern as those of the other experiments.

The relative effectiveness of different CS events is contrasted in *Figure 10A* to show the effect of varying the definition of synchrony further. Comparing the values for the events involving a CS in only one presynaptic PC with those of high synchrony CS events, as defined with a ± 5 ms window, shows the much greater effectiveness of the synchronous CS activity (these are the same values as the left- and right-most points, respectively, on the filled symbol curves in *Figure 7B–E*). As was

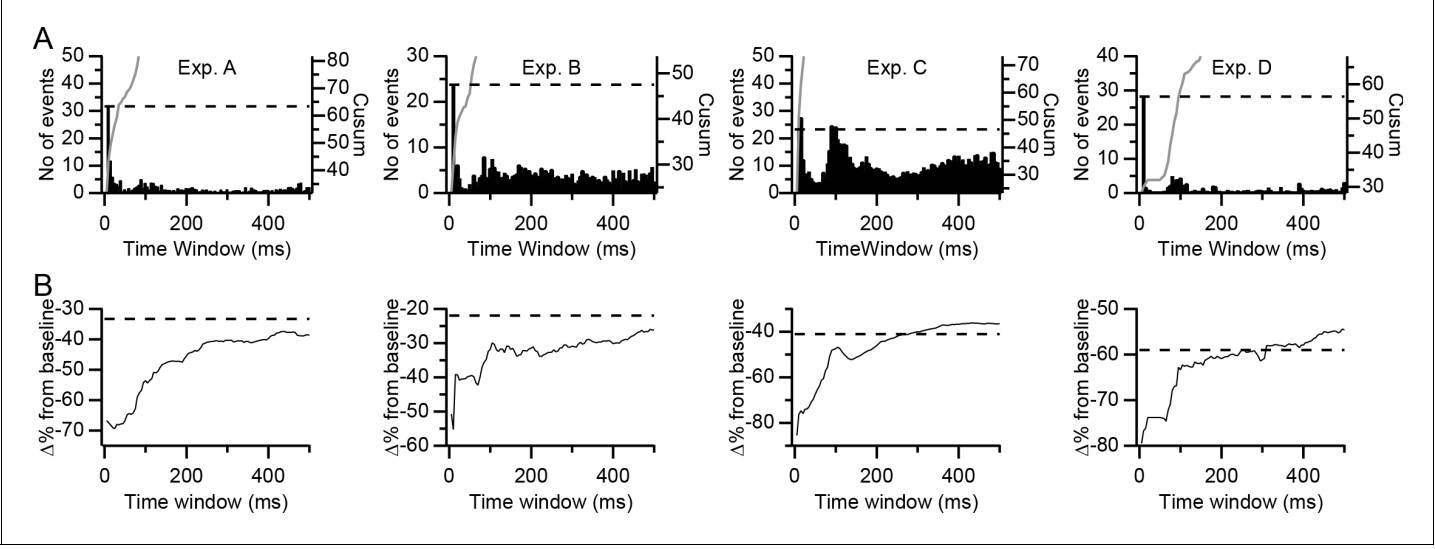

**Figure 9.** Dependence of inhibitory effect on the precision of synchrony. (**A**) Histograms of the number of highly synchronous events among presynaptic PCs as a function of the time window used to define synchrony for each of the experiments. Each histogram bar represents the number of events in that window but not in any shorter window. Each bar reflects a 5 ms increase in window size for both positive and negative latencies from the reference CS. Gray traces are the cumulative summation (cusum) curves. Dashed lines indicate twice the height of the first histogram bar (±5 ms time window) as plotted using the cusum (right) axis. (**B**) Plots of the inhibitory effect (percent change from baseline) on DCN activity of highly synchronous CS events as a function of the time window. Dashed lines indicate the overall inhibitory effect of all CSs in the presynaptic group PCs.

DOI: https://doi.org/10.7554/eLife.40101.010

shown in *Figure 9B*, expanding the window for defining high synchrony events reduces their effectiveness. So, we expanded the time window such that the total number of synchronous events was twice that found in the ±5 ms window (*Figure 10A*, Cusum; note that this is the window at which the cusum curve crosses the dashed line in *Figure 9A*). In three of the four experiments (B, C, and D),

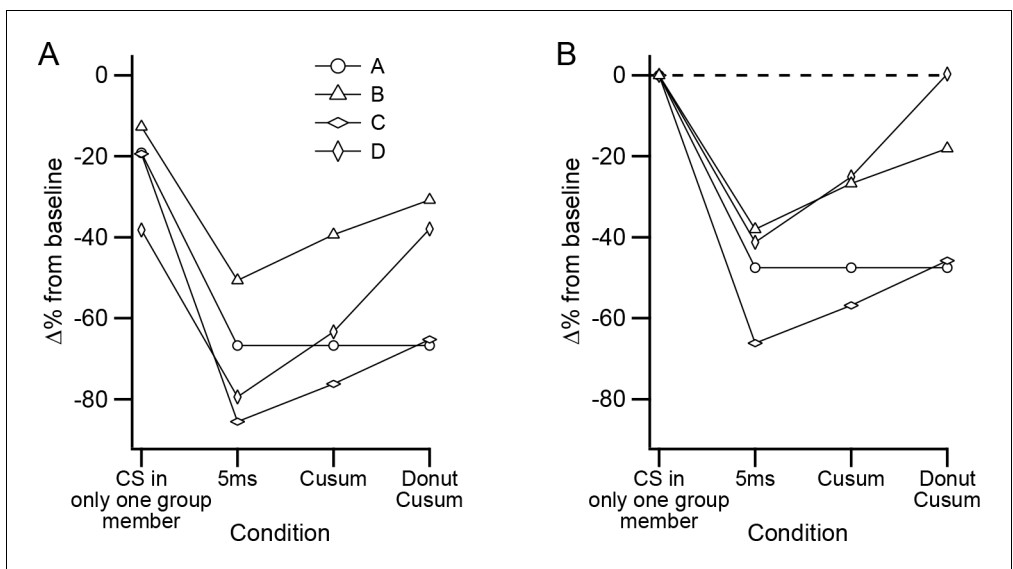

**Figure 10.** Precisely synchronized CS events are responsible for inhibitory effect on DCN activity. (**A**) Plot of different sets of CSs comparing their effectiveness in inhibiting DCN activity. Each symbol shows the data from a separate experiment. (**B**) Data in (**A**) replotted after subtracting the inhibitory effect of events in which only one PC fired a CS.

DOI: https://doi.org/10.7554/eLife.40101.011

the inhibition is reduced compared to that of the ±5 ms window. However, by definition, half of the events in this window are those from the ±5 ms window. Thus, we removed these most precisely synchronized events (i.e., those whose CSs all fell within the central portion of the cusum window) and calculated the inhibition caused by the remaining half of the events in the original Cusum window (*Figure 10A*, Donut cusum). For experiments B, C, and D, this caused a further reduction in the inhibitory effect. Indeed, for experiment D, the inhibition became equivalent to that of events in which only a single PC fired a CS, as shown in *Figure 10B*, where the inhibitory effect has been normalized by subtracting the value of these single PC events. In sum, in most cases, precisely synchronized CS activity causes a stronger inhibition of DCN activity than does more temporally dispersed activity.

## CS-associated, longer latency modulation of DCN activity and CS synchrony

The rasters in *Figure 6* show a slower modulation of DCN activity that is associated with CS activity in addition to the short latency effects just described. This slow, ~1 Hz modulation was prominent in three experiments (exps. A, C, and D) and absent in one (Exp. B). For the experiments that showed the modulation, we investigated whether the increases in DCN activity before and after the CS were related to CS synchrony.

The periods of increased activity before and after the CS were determined from the CS-DCN correlogram generated using all CSs (*Figure 11A*). Here, the baseline was defined as the average level of the histogram between ±1000 ms, and the times at which each period started and ended were determined visually by when the histogram crossed this baseline (pre-CS period: −414 ± 60 ms to −136 ± 65 ms; long latency post-CS period: 146 ± 19 ms to 456 ± 40 ms; n = 3). The average change in activity during these periods relative to the baseline was plotted as a function of CS synchrony, as was done for the short-latency inhibition (*Figure 11B,C*). Although activity in the pre-CS period shows a trend toward increasing with synchrony, the correlation was not significant (r = 0.45, p=0.08, n = 16). However, a significant correlation was found for the long-latency post-CS period (r = 0.56, p=0.024, n = 16).

## Discussion

The present results provide new information about the relationship between PC and DCN activity, and in particular, they highlight the importance of CS activity in shaping cerebellar output. Specifically, the results show that PCs that project to the same DCN neuron are located within the same

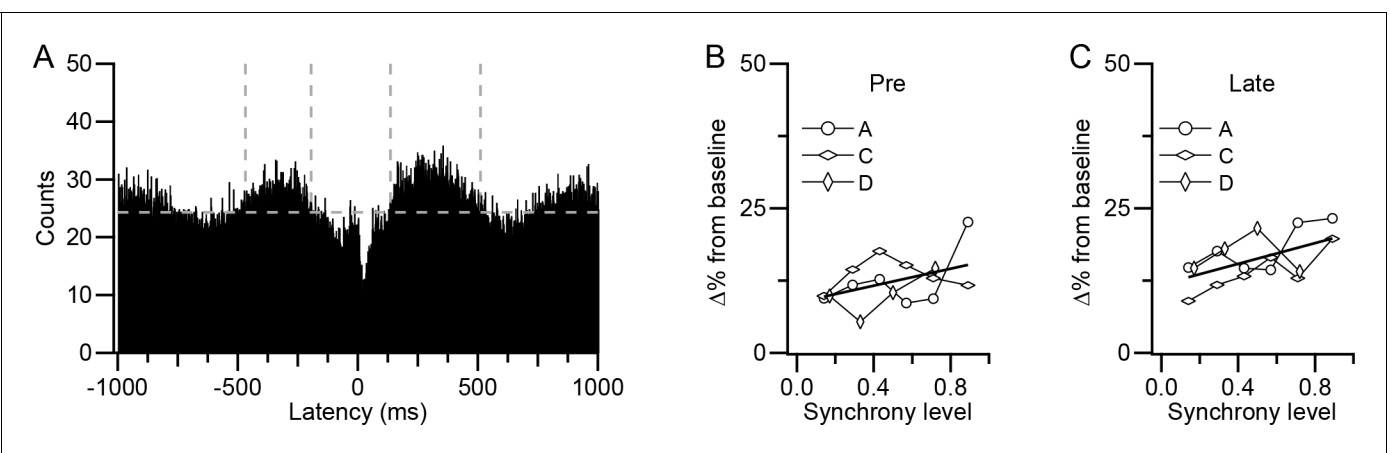

**Figure 11.** Other aspects of CS-associated modulation of DCN activity show weaker relationship to synchrony. (**A**) CS-triggered correlogram of DCN activity generated from all CSs of all presynaptic PCs for experiment A. Horizontal dashed line indicates baseline level of histogram and vertical dashed lines indicate limits of long-latency modulation period. (**B–C**) Change from baseline activity during the pre-CS (**B**) and post-CS (**C**) periods as a function of synchrony. Each curve is the average effect for one experiment. Experiment B did not show clear long-latency modulation and thus was not plotted.
DOI: https://doi.org/10.7554/eLife.40101.012

narrow, rostrocaudally-oriented strip of cortex and have relatively high levels of synchronous CS activity with each other. Moreover, the effectiveness of the CS activity of these PCs in inhibiting DCN activity was shown to be strongly correlated with the level of synchrony. Lastly, the precision of synchrony (i.e., the temporal dispersion of CSs at which their ability to inhibition DCN activity begins to decrease) was found to be ~10 ms in most cases.

## Identification of synaptically-connected PC-DCN cell pairs

The significance of the present findings rests in large part on whether PC-DCN synaptic connections were correctly identified by the analysis of CS-triggered crosscorrelograms of DCN activity. Cross-correlograms are an accepted method for identifying synaptic connectivity (*Perkel et al., 1967*; *Aertsen et al., 1989*; *Csicsvari et al., 1998*). Here, the specific timing of the correlogram event used for identification (i.e., the start of decreased DCN activity at a latency from the onset of the CS that matches the conduction time of the PC-DCN pathway) is consistent with the CSs in the identified PC directly causing this inhibition, but there is still a possibility that this pattern reflects a spurious correlation instead of a true synaptic connection.

However, consideration of the known anatomy of the involved circuits when combined with the present findings on the spatial distribution of the presynaptic groups, provides further support that the correlograms have identified synaptic connections. Specifically, every group of presynaptic PCs had a spatially-restricted distribution in the mediolateral direction, which follows the known topography of the PC-DCN projection in which each DCN region receives input from a narrow rostrocaudally-running cortical strip (*Voogd and Bigaré, 1980*).

In addition, the better maintenance of synchrony with increasing mediolateral separation between PCs in the presynaptic group than in the rest of the PCs in the array is also consistent with their identification as being presynaptic to the same DCN cell, if one considers the relationship between zebrin bands, the topography of the PC-DCN projection, and CS synchrony. First, anatomical studies show that PC input to each DCN region (and hence to each DCN neuron) essentially comes from one zebrin band (*Sugihara and Shinoda, 2007*; *Chung et al., 2009*; *Sugihara, 2011*), so that if the presynaptic group PCs are correctly identified they must all belong to the same zebrin band. Second, PCs within the same zebrin band show high levels of synchrony that are maintained with ML separation, whereas synchrony drops more rapidly with ML separation when PCs are located in neighboring bands (*Sugihara et al., 2007*). Thus, the better maintained synchrony with mediolateral separation for the presynaptic PC groups is consistent with these PCs all being in the same zebrin band, and thus consistent with all of them projecting to the same DCN cell.

It is also worth considering whether DCN afferents other than PCs could have produced an inhibition of DCN activity precisely timed to the onset of a CS, as excluding them would provide further support for the identification process. In this regard, other than PCs, collaterals from the mossy fibers and olivocerebellar axons are the major sources of input to the DCN. They are both excitatory, implying that they would have to show a reduction in activity that is precisely timed to affect the DCN within several milliseconds of a CS. The olivocerebellar system is, by definition, active when a CS occurs, and so reduction in the activity of these collaterals cannot be driving the decrease in DCN activity. It is also implausible that a reduction in mossy fiber activity precisely timed to within a few milliseconds of a CS underlies this reduction.

Another possibility is that changes in SS activity are responsible for the short-latency inhibition that we used to identify PCs as presynaptic to a DCN cell. This possibility can also be excluded, as it is well-known that there is a pause in SSs following a CS, which would be expected to result in an increase in DCN activity, contrary to what was observed. Note that the various types of modulation of SS activity that can follow the SS pause would have latencies that would make it impossible for them to cause this initial inhibition of DCN activity, but they could contribute to the overall response of the DCN to CSs, as is discussed later.

In sum, it seems highly likely that the decrease in DCN activity immediately following the CS is due to CS activity and thus identifies the action of PCs on the DCN cells. However, we need to consider whether this decrease specifically identifies a synaptic connection between the PC and DCN cell being recorded or whether it could instead reflect the connections from other PCs to the DCN cell. This, in turn, requires consideration of two cases. First, olivary axons branch to innervate about 7 PCs in rats (*Schild, 1970*), and so, in theory, the correlogram pattern used to identify PC-DCN connections could reflect the connection to the DCN cell by any one of the PCs contacted by the same

olivary axon as the PC being recorded, in addition to (or instead of) a connection by the recorded PC itself. However, in this case, the correlogram would still be identifying a PC-DCN synaptic connection, and the CSs of the two PCs (the one being recorded and the hypothetical one innervated by another branch of the olivary axon) would be nearly perfectly correlated. Moreover, olivary axon branches are generally confined to the same narrow cortical strip (*Sugihara et al., 2001*), but cf. *Ekerot and Larson, 1982*), and so the finding of a constricted spatial distribution of the presynaptic group along the mediolateral axis would also remain valid. In sum, even if this scenario were to occur, it would have little impact on the interpretation of the results.

The second possibility reflects the fact that CSs are correlated between PCs because of gap junction coupling between inferior olivary (IO) neurons. However, this correlation, while statistically highly significant, is far less than perfect (e.g., r-values are mostly <0.1 when PCs are in the same rostrocaudal band and much less otherwise, see *Figure 4A*). More importantly, the central peak of the crosscorrelogram of CSs in two PCs can be 5–10 ms or wider. Given these characteristics, the inhibition of DCN activity should start before the CS and no precisely timed increase in it should be observed following the CS if it were due to correlated CS activity from other PCs instead of being directly due to the CSs of the PC being recorded. Indeed, correlograms with this pattern of activity did occur often and were excluded.

In sum, the evidence supports interpreting CS-triggered correlograms that show an onset of decreased DCN activity within several milliseconds following the onset of a CS as evidence for a synaptic connection from the recorded PC (or essentially equivalently from a PC receiving input from another branch of the olivary axon that innervates the recorded PC) to the DCN cell.

## Technical limitations

The major limitation of the present study is the low number of DCN cells for which a group of PCs in the recording array could be identified as presynaptic. This limits the generalizability of the conclusions in terms of whether the results apply to all categories of DCN neurons. Nevertheless, that all four DCN cells for which such a group of PCs were identified showed a similar, strong relationship between CS synchrony and the amplitude of the inhibition of DCN activity suggests that this relationship is not rare. It should be noted that the analyses of each of these cases rests on the analysis of spike trains consisting of thousands of CSs (~2500–12,000) and tens of thousands of DCN spikes (~30,000–50,000), and thus rests on a large amount of data.

A second issue that should be addressed is the use of anesthesia, which was required, given the recording methodology and the need to record both PCs and DCN cells for long periods. Differences in cerebellar activity between ketamine anesthetized and awake animals have been reported (e. g., *Schonewille et al., 2006*). This necessitates caution when drawing conclusions from the anesthetized preparation to the awake animal. However, the critical issue is not whether the patterns of activity in the two conditions match exactly, but rather whether the system's operating characteristics in the two states are similar. Indeed, in many ways, PCs and DCN neurons have similar firing characteristics in the two states, though there are some differences, as just mentioned. For example, CS, SS and DCN firing rates are similar in both conditions. In particular, the reported spontaneous firing rates for DCN cells in alert animals vary widely, both within and between studies; overall, rates range from 10 Hz to greater than 100 Hz (e.g., *Thach, 1968*; *Burton and Onoda, 1978*; *Harvey et al., 1979*; *Chapman et al., 1986*; *Fortier et al., 1989*; *Gruart et al., 2000*; *Sarnaik and Raman, 2018*). The data set from our previous report showed a similar range for the overall DCN population, as did the population of DCN cells for which a presynaptic PC was identified (see *Figure 2A,B* of *Blenkinsop and Lang, 2011*).

More importantly, the patterns of CS synchrony are similar (*Lang et al., 1999*), and the PC-DCN synapse is GABAergic, and thus should not be greatly influenced by ketamine, whose primary action is as an NMDA blocker. Note that ketamine has a negligible effect on GABA-A receptors, except those containing the alpha6 subunit (*Hevers et al., 2008*), but such receptors are found only on granule cells in the rat brain (*Wisden et al., 1992*). In sum, under ketamine, PC CS activity is similar to that in the awake rodent, as likely is the efficacy of the PC-DCN synapse.

Finally, several studies have demonstrated that changes in CS synchrony patterns occur during movement (*Welsh et al., 1995*; *Mukamel et al., 2009*), and CS activity has been inferred to influence interpositus firing patterns during conditioned eyeblinks (*Ten Brinke et al., 2017*), suggesting

that synchrony is a physiologically relevant parameter for determining cerebellar output (i.e., DCN activity).

## Synchrony may be needed to modulate DCN activity

Our results indicate that CSs need to be synchronized to be able to affect DCN activity. They are consistent with previous work showing that the tonic SS activity of individual PCs has a 'near negligible' effect on DCN cells (*Bengtsson et al., 2011*), and with in vitro studies that suggest that DCN firing is more effectively phase locked by synchronized PC activity (*Person and Raman, 2012*; *Person and Raman, 2011*). Nevertheless, it is possible that the CS activity, unlike the SS activity, of an individual PC could significantly influence DCN firing, because a CS often triggers a high frequency burst of axonal spikes (*Ito and Simpson, 1971*; *Khaliq and Raman, 2005*; *Monsivais et al., 2005*). However, at best, this mechanism appears to produce a minimal effect, based on our extrapolation of the relationship between synchronous CS activity and its inhibitory effect on DCN firing down to the effect caused by an individual PC firing a CS (*Figure 7*).

The inability of asynchronous activity from an individual PC to influence the DCN cells it contacts is understandable given the high firing rates of PCs and the convergence of at least dozens of PCs onto each DCN neuron. The need for synchrony is also consistent with the electrophysiology of most classes of DCN cells and the characteristics of their synaptic input. In particular, under in vitro conditions, the membrane time constant is reported to be ~35 ms for nonGABAergic DCN cells (*Uusisaari et al., 2007*); however, most DCN cells are subjected to an intense ongoing barrage of synaptic activity from the dozens of PCs that converge onto each one, which would reduce this value. Indeed, in decerebrate cats, a value of ~7 ms was reported (*Bengtsson et al., 2011*). Furthermore, the GABAergic IPSCs of most classes of DCN cells are extremely brief, lasting only ~2 ms on average (*Person and Raman, 2011*). In sum, the presynaptic and intrinsic properties of most classes of DCN cells suggest that they should function as coincidence detectors, consistent with our finding that precisely synchronized (within 5–10 ms) CSs are most effective in inhibiting DCN activity.

Although the above discussion generally applies to most classes of DCN neurons, there are exceptions. For example, DCN neurons that project to the IO have very long IPSC time constants (*Najac and Raman, 2015*), and likely have longer membrane time constants than most other DCN cells as well (*Uusisaari et al., 2007*). Thus, modulation of the activity of these cells would not be expected to require precisely synchronized synaptic input. In this regard, it is worth noting that one experiment (Exp. A) was distinct from the others, in that far less precise CS synchrony was needed to inhibit DCN activity strongly. Indeed, in this case, CSs dispersed over about 50 ms still produced as strong DCN inhibition as precisely synchronized CSs. It is tempting to speculate that in this case we were recording a DCN-IO projection neuron. Consistent with this, the firing rate of the DCN cell in Exp. A was 29.6 Hz, which is relatively low for a DCN neuron in vivo and consistent with the lower firing rates reported for DCN-IO neurons, and small GABAergic DCN neurons generally, recorded in vitro (*Uusisaari et al., 2007*; *Najac and Raman, 2015*). However, additional experiments with larger numbers of DCN recordings and where IO projecting neurons are identified as such (e.g., by antidromic stimulation) are needed to address this possibility.

## CS synchrony: direct and indirect actions on the DCN

Exactly how synchronous CS activity translates into changes in DCN activity is not straightforward, in part because CSs are associated with changes in SS activity and therefore could influence the DCN both directly, because of the axonal spikes triggered by the CS, and indirectly, via changes in SS activity. The latter includes the pause in SSs following a CS and the further modulation of SS activity following the pause (*Granit and Phillips, 1956*; *Bell and Grimm, 1969*; *Murphy and Sabah, 1970*; *Bloedel and Roberts, 1971*; *Latham and Paul, 1971*; *Burg and Rubia, 1972*; *Ebner and Bloedel, 1981a*; *Ebner and Bloedel, 1981b*; *McDevitt et al., 1982*; *Bloedel et al., 1983*; *Ebner et al., 1983*; *Mano et al., 1986*; *Sato et al., 1992*; *Burroughs et al., 2017*; *Tang et al., 2017*). Moreover, these direct and indirect effects are not necessarily mutually exclusive, and indeed, we would argue that both are likely needed to explain fully the modulation of DCN activity we observed.

To begin, the short-latency inhibition indicates that a direct effect of CS activity dominates the initial response of the DCN cell following a CS. In contrast, as mentioned earlier, if the pause in SSs, which occurs during the initial 20–30 ms of the short-latency inhibition, were the dominant

mechanism during this period, the resulting disinhibition would have caused an increase in DCN activity, contrary to what was observed. Moreover, a positive correlation between synchrony and DCN activity, if anything, should have been present rather than the strong negative correlation that was observed. Thus, at least the initial portion of the short-latency inhibition appears to be due to the axonal spikes triggered by the CSs.

The duration of the short-latency inhibition (50–100 ms), however, raises the question of whether it can be attributed solely to the direct effect of CS activity, given the short time course of GABAergic IPSCs and IPSPs in DCN neurons (*Person and Raman, 2011*). The direct effect of the CS, however, is likely to be prolonged by several factors. A CS can have as many as eight spikelets in addition to its initial spike, and they occur over a period of ~20 ms (*Tang et al., 2017*). Although not all spikelets are associated with axonal spikes (*Ito and Simpson, 1971*; *Khaliq and Raman, 2005*; *Monsivais et al., 2005*), the extended nature of this activity means that a DCN cell could be subjected to recurring IPSCs for a period of ~20 ms. Furthermore, we need to consider the precision of the synchrony, as the width of the central peak in CS-CS correlograms typically is on the order of 5–10 ms, and in the current analysis, the narrowest time window (±5 ms) allowed a scatter of up to 10 ms. Combining these factors suggests that a synchronous CS event might lead to IPSCs bombarding a DCN cell over some 30 ms, albeit with decreasing numbers toward the ends of this range. In addition, although the strongest inhibition would occur while the IPSCs were ongoing, the longer duration of the IPSPs would lengthen the inhibitory period for perhaps another 5–10 ms, giving a total duration of ~40 ms, which is close to that of the lower limit of the short-latency inhibition. It is also worth noting that DCN neurons have plateau potentials and can exhibit bistability (*Jahnsen, 1986*; *Llinás and Mühlethaler, 1988*; *Han et al., 2014*), and thus the powerful inhibition caused by synchronous CS activity could shift the DCN cell to a less excitable state that exceeds the duration of the IPSPs.

Even taking all of these factors into account, the duration of the short-latency inhibition appears to exceed what is easily explained by a direct action of CS activity. Thus, we propose that the later stages of the short-latency inhibition of DCN activity also reflect an indirect effect of CS activity to synchronize the modulation of SS activity. Specifically, following its post-CS pause, SS activity often rebounds to rates that are significantly higher than baseline for 50–100 ms (*Tang et al., 2017*). This period of increased SS activity coincides with the second half of the short-latency inhibition period and would thus contribute to maintaining the inhibition. This rebound period is followed by a longer period of mildly decreased SS activity that lasts until ~600 ms after the CS (*Tang et al., 2017*), a period that is similar to that of the long-latency excitation in DCN cells. Moreover, as was the case for the short-latency inhibition, the amplitude of the post-CS long-latency excitation was correlated with the level of CS synchrony, as it should be if it depends on synchronization of PC activity.

## CS synchrony may be key to modulating cerebellar output

The present results combined with those of previous studies (*Bengtsson et al., 2011*; *Person and Raman, 2012*; *Person and Raman, 2011*) suggest that synchrony is a critical parameter of PC influence on the DCN. This in turn points to a central role for the olivocerebellar system in the control of cerebellar output, as it is organized to generate synchronous CS activity across large and varied ensembles of PCs (*Sasaki et al., 1989*; *Sugihara et al., 1993*; *Lang et al., 1996*; *Lang, 2001*; *Lang, 2002*). By contrast, spontaneous SS activity shows comparatively low levels of synchronization between small numbers of neighboring PCs (*Bell and Grimm, 1969*; *Heck et al., 2007*; *Wise et al., 2010*). Moreover, the most obvious mechanism for generating synchronous SS activity (that doesn't involve the olivocerebellar system), shared parallel fiber input, would induce synchronous activity among relatively few PCs that would converge to the same DCN cell, given the spatial alignment of the parallel fibers. In contrast, the olivocerebellar system causes synchronous CS activity in rostrocaudal strips that can extend across multiple cerebellar lobules (*Sugihara et al., 1993*; *De Zeeuw et al., 1996*; *Yamamoto et al., 2001*), and thus among PCs likely to converge onto the same DCN cells. That olivocerebellar synchrony may be the key control parameter of DCN activity does not, however, exclude a role for SSs. Rather, as discussed above, synchronous CS activity may act both directly and by synchronizing the modulation of SS activity.

## Materials and methods

Experiments were performed in accordance with the National Institute of Health *Guide for the Care and Use of Laboratory Animals*. Experimental protocols were approved by the Institutional Animal Care and Use Committee of New York University School of Medicine. All surgical and recording procedures were performed under general anesthesia, and every effort was made to minimize suffering.

### General surgical and recording procedures

Female Sprague-Dawley rats (225–300 g) were initially anesthetized with ketamine (100 mg/kg, ip) and xylazine (8 mg/kg, ip). Supplemental anesthetic was supplied continuously via a femoral catheter to maintain a constant depth of anesthesia. The depth of anesthesia was assessed by paw pinch and by the absence of spontaneous movements. Rectal temperature was maintained at 37°C using a heating pad connected to a temperature control system. To gain access to the cerebellum, animals were placed in a stereotaxic frame, and a craniotomy was performed to expose the posterior lobe of the cerebellum. The overlying dura mater was then removed, and the cortical surface was stabilized and protected by covering it with a platform constructed from an electron microscope grid that was pre-embedded in a thin sheet of silicone rubber and supported by tungsten rods. Gelfoam pieces soaked in Ringers solution were used to cover any remaining brain surfaces. Dental cement was then used to fix the platform to the skull and to seal the craniotomy.

Extracellular recordings of CS activity were made using a multiple electrode technique in which glass microelectrodes (50/50 mixture of 2M NaCl solution and glycerol) were implanted by driving them through the rubber of the spaces in the grid using a micromanipulator. Electrodes were implanted into the molecular layer to a depth of 100–150 µm below the folial surface, where CSs but not SSs can be observed. To record SSs, the electrodes need to be implanted to a depth of ~225–250 µm, close to the PC layer in rats. While also obtaining SS recordings is desirable, there are several reasons why this was not done. These include a much longer implantation process, greater overall damage to the cortex, more difficult spike sorting for analyses, and less stability during recording (i.e., the electrode is much more likely to damage or kill the cell being recorded because of its closer proximity to the cell body).

Electrodes were implanted with a spacing of ~250 µm using the rows and columns of the electron microscope grid as a guide. For further details on the multielectrode technique, see (*Sasaki et al., 1989*; *Lang, 2018*). Once an array of electrodes was implanted to crus 2a, a single glass microelectrode (2M NaCl) was lowered under stereotaxic guidance through either the crus 1b or the paramedian lobule to the DCN, where recordings of well-isolated single units were obtained. Upon good isolation of a DCN cell, the activity was typically recorded for 20 min. At the end of the recording session, the animal was perfused under deep anesthesia and the cerebellum was removed for histological processing.

All neuronal activity was recorded using a multichannel recording system (MultiChannel Systems MCS GmbH, Reutlingen, Germany) with a 25 kHz/channel sampling rate, gain of 1000x, and band pass filters set at 0.2–8.0 kHz. Spike sorting of CSs and DCN activity was done offline using routines written in Igor Pro (Wavemetrics, Lake Oswego, OR, USA).

### Data analysis

In the text, all mean values are given ±one standard deviation. Populations were tested for normality using the Kolmogorov-Smirnov (KS) goodness-of-fit test. For normally distributed populations, paired or unpaired t-tests, as appropriate, were used to compare means. When differently sized populations were compared, t-values and degrees of freedom were calculated using formulas that account for differences in population size (Welch's approximate t) (*Zar, 1999*). For non-normally distributed populations a nonparametric test was used and is indicated in the text. Correlation coefficients (Pearson's r) were tested for significance using the formulas given in (*Zar, 1999*).

#### Identification of monosynaptically-connected PC-DCN cell pairs

CS-triggered histograms of DCN activity were used to identify PCs that had a monosynaptic connection to the DCN cell being recorded, as described previously (*Blenkinsop and Lang, 2011*). Identification of a PC-DCN connection rested on the existence of a significant negative deflection in the CS-triggered correlograms within 5 ms after CS onset, consistent with the conduction time of the

PC-DCN pathway and inhibitory nature of PCs. Significance was assessed by testing whether the slopes of the histogram just prior and following the CS were different (*Zar, 1999*). Specifically, the period preceding the CS was −10 to 0 ms in almost all cases, but was −15 to −5 ms if an excitatory response was visible in the histogram at the time of the CS. The post-CS period was 3 to 13 ms. On this basis all PCs in a recording array were divided into presynaptic and non-synaptic groups (i.e., PCs that were connected or were not connected to the DCN cell being recorded).

### Synchrony analysis

To determine the average CS synchrony level of a pair of PCs for the entire recording session, a cross-correlation coefficient was calculated by treating the spike trains as zero-one variables (*Gerstein and Kiang, 1960*; *Sasaki et al., 1989*). The spike train of a cell was represented by X(i), where i represented the time step (i = 1, 2,. .., N). X(i)=1 if a CS onset occurred in the ith time bin, otherwise X(i)=0. Y(i) represented the spike train of the reference cell and was defined the same as X(i). The synchrony level, C(0), was then calculated as

$$C(0) = \left[\sum_{i=1}^{N} V(i) * W(i)\right] / \sqrt{\sum_{i=1}^{N} V(i)^2 * \sum_{i=1}^{N} W(i)^2}$$

where V(i) and W(i) were

$$V(i) = X(i) - \sum_{j=1}^{N} \frac{X(j)}{N}, \; W(i) = Y(i) - \sum_{j=1}^{N} \frac{Y(j)}{N}$$

Note that C(0) ranges between ±1, having a value of 1 when the two spike trains are perfectly synchronized and a value of 0 when the spikes of the two cells are independent of each other. In previous work we have used simulated, randomized and time-reversed spike trains to assess the statistical significance of the experimentally-observed C(0) values and in all cases the values for the experimental data are well above the 99th percentiles for the various test data (*Sugihara et al., 1993*; *Lang et al., 1996*).

It was also necessary to determine the synchronization level in the presynaptic group at the times of individual CSs in each PC of the group. For this purpose, each PC served as the reference cell, and the time of the onset of each of its CSs marked the center of a time window that defined the event. The synchrony level for each such event was then defined as the fraction of PCs in the presynaptic group that fired CSs within the specific time window. A ± 5 ms window was used except where stated otherwise. Thus, all CSs could be assigned a synchrony level ranging between 1/x (no CSs occurred in other PCs) and 1 (all PCs fired a CS within the time window), where x is the number of PCs in the presynaptic group.

## Acknowledgements

This work was supported by grants to EJL from the National Science Foundation (IOS-1051858) and the National Institute of Neurological Disorders (NS-37028; NS-95089).

## Additional information

### Funding

| Funder | Grant reference number | Author |
|---|---|---|
| National Institute of Neurological Disorders and Stroke | NS-37028 | Eric J Lang |
| National Institute of Neurological Disorders and Stroke | NS-95089 | Eric J Lang |
| National Science Foundation | IOS-1051858 | Eric J Lang |

The funders had no role in study design, data collection and interpretation, or the decision to submit the work for publication.

## Author contributions

Tianyu Tang, Formal analysis, Writing—original draft, Writing—review and editing; Timothy A Blenkinsop, Formal analysis, Investigation, Methodology, Writing—review and editing; Eric J Lang, Conceptualization, Resources, Software, Formal analysis, Supervision, Funding acquisition, Investigation, Methodology, Writing—original draft, Project administration, Writing—review and editing

## Author ORCIDs

Eric J Lang (iD) http://orcid.org/0000-0002-8524-9854

## Ethics

Animal experimentation: Experiments were performed in accordance with the National Institute of Health Guide for the Care and Use of Laboratory Animals. Experimental protocols (030202-02, 030202-03) were approved by the Institutional Animal Care and Use Committee of New York University School of Medicine. All surgical and recording procedures were performed under general anesthesia, and every effort was made to minimize suffering.

## Decision letter and Author response

Decision letter https://doi.org/10.7554/eLife.40101.017
Author response https://doi.org/10.7554/eLife.40101.018

# Additional files

## Supplementary files

• Transparent reporting form
DOI: https://doi.org/10.7554/eLife.40101.013

## Data availability

The spike train data has been deposited in Dryad at doi:10.5061/dryad.61pt552

The following dataset was generated:

| Author(s) | Year | Dataset title | Dataset URL | Database and Identifier |
|---|---|---|---|---|
| Tang T, Blenkinsop TA, Lang EJ | 2018 | Data from: Complex spike synchrony dependent modulation of rat deep cerebellar nuclear activity | https://dx.doi.org/10.5061/dryad.61pt552 | Dryad Digital Repository, 10.5061/dryad.61pt552 |

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
