## [Decision Letter]

[**Editorial note:** This article has been through an editorial process in which the authors decide how to respond to the issues raised during peer review. The Reviewing Editor's assessment is that all the issues have been addressed.]

Thank you for submitting your article "Complex spike synchrony dependent modulation of rat deep cerebellar nuclear activity" for consideration by *eLife*. Your article has been reviewed by three peer reviewers, including Vatsala Thirumalai as the Reviewing Editor and Reviewer #1, and the evaluation has been overseen by Richard Ivry as the Senior Editor. The following individual involved in review of your submission has agreed to reveal her identity: Chris I De Zeeuw (Reviewer #3). Reviewer #2 remains anonymous.

The Reviewing Editor has highlighted the concerns that require revision and/or responses, and we have included the separate reviews below for your consideration. If you have any questions, please do not hesitate to contact us.

Summary:

This manuscript presents a nice meta-analysis of synaptically connected Purkinje neurons and deep cerebellar nuclear cells. Importantly, it presents the idea that inhibition of spiking in nuclear cells is controlled by the synchrony of complex spikes in the presynaptic Purkinje neuron population. All three reviewers agreed that this is a significant finding and appreciate the authors for this insight. However, all three reviewers also had concerns that need to be addressed and suggestions for improving the manuscript.

Major concerns:

One of the concerns is the low number of DCN cells recorded from, especially because of the heterogeneity of DCN cells. While the efforts required to find PC-DCN pairs are akin to finding a needle in a haystack (x4), it is important that the authors address the weakness of low n's in their discussion of the results. There are also some other caveats (such as indirect synaptic connections, the use of anaesthesia, etc.,), which should be addressed by the authors in the discussion.

In general, reviewers felt that the manuscript could be rewritten such that results are presented thoughtfully and even-handedly, with an awareness of the strengths and limitations that is conveyed to the expert and non-expert reader. Reviewers also had several suggestions to improve the presentation of figures and overall lucidity of the manuscript and these specific suggestions can be found in the individual reviews appended below.

Separate reviews (please respond to each point):

*Reviewer #1:*

This manuscript argues that the synchrony of complex spikes in the Purkinje neuron cohort projecting to the same DCN neuron might hold the key for controlling cerebellar output. The manuscript is written in clear style and the data are analyzed and presented well. My biggest worry is that the data presented are from only four or five DCN cells (Results paragraph one says five cells, but four experiments are presented. Authors need to clarify how data from two DCN cells in the same animal were combined). In their dataset, authors could identify 4-7 PCs to be synaptically connected to the DCN cell being recorded. This combined with the low number of highly synchronous CS inputs in each experiment (Figure 6B), places the findings on not-so-solid ground. In addition, of the four, one DCN cell (A) did not show the tight synchrony-dependent modulation seen in the other three cells (Figure 9B and 10). The authors speculate that this cell could be an IO projecting DCN but this idea is not tested. With this in mind, I am not certain that the main point the authors push, i.e. that CS synchrony in a 10ms window is the most important parameter for DCN output, can be accepted. Some specific points:

1) Figure 1 is not useful. Either remove or include PC recordings and a Venn diagram of cell numbers discussed at the end of paragraph one of the Results.

2) Will be useful to indicate the long latency modulations of DCN activity in Figure 2, with an arrow or grey bar.

3) In Figure 3, authors show lines to indicate 95th and 99th percentile ML separation and state that the synaptic group value is above the 99th percentile. This should be the 5th and 1st percentile and "below the 1st percentile" given that the synaptic group is placed more closely than the others.

4) Figure 4A: SD or SEM? Prefer that the authors would show the individual synchrony values of all PC's in that experiment as scatter with an indication of the mean. Similar plots can be made for all four experiments. Figure 4B does not make much sense to me. What does it mean that the ratio is several-fold over very long distances?

5) Figure 11B: Authors argue that the long latency modulation of DCN activity is also synchrony dependent but the effect is weak and variable. One of the four cells recorded did not show this modulation. This point appears weak and not related to the main point of the manuscript.

6) Results paragraph two, final sentence: What is the larger sample used in this comparison?

7) Subsection “Inhibition of DCN activity varies with level of CS synchrony” paragraph five: insert "Figure 7F"

8) Sixth paragraph of the same section: Could singular CS's still be high synchrony events synchronized with PCs not being recorded from?

9) Subsection “Synchrony may be needed to modulate DCN activity”: "Our results indicate that CSs need to be synchronized to be able to affect DCN activity". Given that singular CSs also show about 20% inhibition, one would argue that they are also able to affect DCN activity, albeit at a lower level relative to the synchronous ones. Again, as noted above, it is likely that the singular CSs are in fact a mixed population of events of widely varying synchronicity. This point needs to be brought out in the discussion.

10) Subsection “CS synchrony: direct and indirect actions on the DCN” paragraph four: later

*Reviewer #2:*

This manuscript is a meta-analysis of multielectrode data from recordings of Purkinje cells (PCs) and deep cerebellar nuclei (DCN) in anesthetized rats. The authors correlate activity of PC complex spikes and DCN spikes to find likely connected cells and then explore the effects of synchronized PC complex spikes on 4 DCN cells. The greater the degree of complex spike synchrony, the greater the inhibition of DCN cells. The manuscript has useful information and the authors are careful to report all the exceptions and special cases, but the text doesn't always make it clear what the relevant points are or what aspects of the data the authors wish to stress. There are relatively few points along the way through 11 figures where the results and interpretations are synthesized. It would be helpful if the text were re-written to include a few more summary sentences. Likewise, there are many figures with relatively few panels, but sometimes the transformations of the raw data aren't immediately obvious. Possibly some further illustration would be helpful. Specifics are given below.

Comments.

1) Figure 1 has example traces of 4 DCN cells, but nothing else, and it is hard to know what we are looking at them for. Are these the 4 cells that are identified in Figure 3 and carried through the paper? Figure 2 also shows 4 histograms from DCN cells but, in this case, 2 plots are from one DCN cell. PC spikes and PC synchrony are never actually illustrated. It could be helpful to see some of the PC traces (and maybe combine Figures 1 and 2 and relate them to 3) so that the raw data are clearer.

2) Figure 3 is used to make the argument that the likelihood of finding converging PCs decreases with mediolateral (M-L) but not rostrocaudal (R-C) distance. This argument is convincing in the M-L dimension but less so in the R-C dimension, since the multielectrode array has 7-9 electrodes M-L but only 3-4 R-C, and in Figure 3E and F the converging PCs have about equivalent mean M-L and R-C separations. Should this analysis be curtailed to talk about clustered PCs rather that PCs in an R-C plane?

3) The Synchrony Index described in the Materials and methods and used first in Figure 4 could be explained more clearly and intuitively, possibly even illustrated. The equation indicates that it is the summed product of time-matched deviations from the mean for each of two cells, divided by the root of the product of the summed-square-deviations. It is clear that when both signals go high simultaneously (i.e., synchronize), The Synchrony Index value will increase, but it is not immediately obvious what the range of values is expected to be, how this will vary with different levels of background firing, etc. It would be useful if the authors could help calibrate the reader, so that the values of Figure 4 (0.12, 0.06) can be placed in perspective. This figure might be a useful place to illustrate the successive transformations of the data.

4) Subsection “Inhibition of DCN activity varies with level of CS synchrony”; “Although singular CSs had the weakest inhibitory effect, they still reflect a significant level of synchronization….” It is not clear on what basis the authors say that the “singular” CSs, which seems to mean non-synchronized CSs, reflect synchrony. The argument about the 50-100 converging PCs is hard to follow, especially when the concluding sentence indicates that the effect is not different from no synchrony. Please clarify.

5) As far as this reader can tell, the central, most important finding of the manuscript is Figure 7F, which shows that for all four DCN cells, there is a strong correlation with steep slope of the amount of inhibition as a function of synchrony level. This key panel is not called in the text and hardly stands out in the illustrations. With so many figures and a relatively monotonic (without emphasis) presentation, it is possible that it will be missed. Some rewriting would help make this result more salient.

6) The 7F panel is complemented by Figure 8. It switches to illustrating the inhibitory efficacy of PCs based on whether they are in a synchronous group, but the authors just refer to this as a “cell-by-cell” comparison, so it takes several readings before one realizes that this is a demonstration from the “perspective” of the PCs, which is quite important. Please rephrase for clarity. (Perhaps the authors could consider putting this key panel with Figure 7F somehow so that they stand out as a central figure?)

7) Discussion; “these [convergent] PCs have highly synchronized CS activity…” Is what is meant here that they have a high probability of synchronizing (which does not appear to have been tested; or was it?) or that when they synchronize they do so with <5 ms jitter (which is stated again in the last clause)?

Minor points.

1) Results; “In a few cases (n=4)” Please indicate n=4 out of how many cases.

2) Figure 2, please indicate bin width in legend.

3) Figure 4B, perhaps different line patterns (solid, dashed, dotted) could help distinguish the four curves.

4) The use of the word “singular” to describe non-synchronous CS's is rather non-standard and unintuitive. Perhaps a definition could be given at first use.

5) Figure 5, it would be useful to indicate the number of PCs in each group in the legend or on the plots.

6) Subsection “Inhibition of DCN activity varies with level of CS synchrony”; (Figure 7) “the baseline activity in all histograms was normalized for sweep number…” It is not clear what analysis was done. Please clarify.

7) Please define “cusum.”

8) The word “axonic” should probably be “axonal.”

*Reviewer #3:*

The authors of the manuscript entitled "Complex spike synchrony dependent modulation of rat deep cerebellar nuclear activity" investigated how Purkinje cell complex spike activity and synchrony affects activity in their main downstream target, the cerebellar nuclei (CN). By doing simultaneously recordings of Purkinje cells (PCs) in the cerebellar cortex and of CN neurons, they describe that PCs that are putatively presynaptic to recorded CN cells are located in rostrocaudal strips. In addition, they describe that CN cells show specific patterns of short-latency responses to complex spike activity of which the strength depended on the degree of complex spike synchrony. The paper addresses an important and timely topic using a series of detailed in depth analyses.

I have several suggestions to improve the paper.

1) The manuscript largely lacks raw data. Only in Figure 1, traces are shown but at a very compressed time scale. Please expand.

2) The number of CN cells from which they recorded in a paired format with PCs is low (n = 4/5). Given that there are many different cell types in the nuclei (e.g., Dulac and Uusisaari studies), it may be worthwhile to increase the number of CN cells. In addition, the number of high-synchrony events per cell is low. This makes the conclusions, although likely to be accurate, somewhat preliminary.

3) The authors discriminate between directly coupled Purkinje cell – DCN neuron pairs and non-synaptic pairs. Their criteria for establishing synaptic connections are not mentioned in the current paper, but a reference is made to a previous publication (Blenkinsop and Lang, 2011). Also in that publication, the criteria are not overly clear. In the latter study, the impact of indirectly coupled pairs has also been described (see Figure 4C of Blenkinsop and Lang, 2011). As the authors discuss (subsection “Identification of synaptically-connected PC-DCN cell pairs”), identification of synaptic connections is difficult and could be compromised by synchrony between Purkinje cells. As Purkinje cells with a high level of synchrony might be considered functionally equivalent, the authors rightfully remark that it is questionable whether this would affect their conclusions. However, the abovementioned, indirect connections (see Figure 4C of Blenkisop and Lang, 2011), could also affect the patterns. Thus, it is crucial to understand which criteria were used to define mono-synaptic connections and to what extent indirect connections could contribute to the findings.

4) Regarding the indirect connections: the authors focus on the synaptically connected pairs, but was there also an impact of complex spike firing in the non-synaptic group? According to Figure 4A (of the present manuscript), also these cells do show synchronous complex spike firing, albeit at a lower level, so that an effect would be predicted. Please discuss.

5) The experiments were performed in anesthetized animals (with ketamine/xylazine). Since this anesthesia can affect cerebellar firing patterns including patterns of rhythmicity and synchrony (e.g., Schonewille et al., 2006), it will be worthwhile to replicate some recordings in awake behaving animals. This will increase the functional relevance of the study.

6) In addition, or alternatively, the authors may want to discuss the Ten Brinke studies (2015 and 2017) in which complex spikes and CN activity were recorded during eyeblink conditioning. In these studies the same patterns were detected with separated single unit recordings of PCs and CN cells in awake behaving animals as described with simultaneous dual recordings in the current study. This would highlight the achievement of the current dual recordings of the current Lang study, while clarifying the putative relevance of it during natural behavior. In fact, the CS synchrony and impact thereof in the CN appear to be enhanced during motor learning.

7) The authors relied on 1 ms binning of their data. The data underlying Figure 7 depend on the duration of the inhibition, which is defined as ending when the 3-point average reaches baseline levels again. This seems a very noise-prone analysis, especially for relatively rare occurrences of high-synchrony (see Figure 6). The open symbols in Figure 7B-E are therefore questionable; the closed ones a bit more reliable, but even here an estimate of the baseline firing during only 50 or 25 ms (the choice of either which value being not really justified), might be quite arbitrarily. Overall, the conclusion that more synchrony leads to more inhibition seems plausible, but the statistical evaluation could be improved (see also below). One should notice, again, that all these conclusions are based upon only four cells.

Additional data files and statistical comments:

The authors report using paired and unpaired t-tests in addition to Pearson correlation analyses. However, their reasoning for choosing these tests is unclear. Based on the low n, inequality of sample sizes (e.g. the difference in number of low vs high synchrony events) and questionable normality of distributions, one may also have to consider non-parametric testing; this reviewer is not a statistician, but please ask advice from a professional to make sure the current approach is correct or not. For sure, the authors should include a detailed description of tests used (and for what reasons they were used) and also include the specific outcomes like t-values, degrees of freedom etc. Evaluation of the statistics may also indicate how many pairs of PC-CN cells need to be illustrated in the current paper to back up the conclusions. So this reviewer has no real doubts about the conclusions of the current manuscript, but they need to be properly backed with solid statistics to stand the test of time.

[Editors' note: further revisions were requested prior to acceptance, as described below.]

Thank you for resubmitting your article "Complex spike synchrony dependent modulation of rat deep cerebellar nuclear activity" for consideration by *eLife*. Your article has been reviewed by three peer reviewers, including Vatsala Thirumalai as the Reviewing Editor and Reviewer #1, and the evaluation has been overseen by Richard Ivry as the Senior Editor. The following individual involved in review of your submission has agreed to reveal his identity: Chris I De Zeeuw (Reviewer #3).

The manuscript is indeed written more accessibly than in the previous submission, but there are some remaining issues that we feel should be addressed before your paper is published, as outlined below:

1) The authors claim (subsection “Technical limitations”) that DCN firing rates are the same with and without anesthesia. No citation is given. The 34 Hz DCN firing rate seen here seems relatively low. The main line of the authors' arguments doesn't rely on claiming that there is no effect of anesthesia, and so unless there is explicit evidence, it is probably better not to say that anesthesia does not affect the firing rates.

2) Figure 1: Reviewers wanted PC recordings to be shown here and this has not been done or addressed in the response.

3) Figure 4A: Individual PC synchrony values – not been done or addressed in the response (reviewer #1, point 4).

4) Figure 4: No illustration/schematic for data transformation used in this and subsequent figures, as suggested by reviewer 2.

5) No response provided for Point #6 from reviewer 2.

6) Figure 4: No statistical tests seem to have been done?

7) Figure 5: Will be better to plot the different lines in different colors, especially for Figure 5E where the spread around the mean can also be shown.

Results paragraph three: Were SS firing rates compared?

Subsection “Inhibition of DCN activity varies with level of CS synchrony” paragraph nine and elsewhere: Wherever p values are given, please state the n's and the test used.

Subsection “What is the effect of non-synchronized CS activity?” first paragraph: "This is not…" Odd sentence construction

Subsection “Identification of synaptically-connected PC-DCN cell pairs” and response letter: Authors argue that earlier work (Sugihara, 2007) shows that CS synchrony is higher within a zebrin compartment than between. However, that paper showed that this was true only for central and lateral crus2a. In medial crus2a, wide synchrony bands covering more than one zebrin compartment were observed. In the present study, it is not clear from which region of crus 2a neurons were recorded from.

Subsection “Identification of synaptically-connected PC-DCN cell pairs” paragraph five and elsewhere in the Discussion: "…as it is well-known that there is a pause in SSs following a CS," – A CS may have diverse effects on SS firing such as facilitation, pausing or suppression (De Zeeuw et al., 2011) or state switches (Loewenstein et al., 2005; Sengupta and Thirumalai, 2015). In this and other places in the manuscript, the writing seems to suggest that CS leads only to SS pauses.

---

## [Author Response]

Major concerns:One of the concerns is the low number of DCN cells recorded from, especially because of the heterogeneity of DCN cells. While the efforts required to find PC-DCN pairs are akin to finding a needle in a haystack (x4), it is important that the authors address the weakness of low n's in their discussion of the results. There are also some other caveats (such as indirect synaptic connections, the use of anaesthesia, etc.,), which should be addressed by the authors in the discussion.In general, reviewers felt that the manuscript could be rewritten such that results are presented thoughtfully and even-handedly, with an awareness of the strengths and limitations that is conveyed to the expert and non-expert reader. Reviewers also had several suggestions to improve the presentation of figures and overall lucidity of the manuscript and these specific suggestions can be found in the individual reviews appended below.

The major concern was that the discussion should address the low n's regarding the number of DCN cells in the discussion as well as other caveats such as the use of anesthesia and indirect synaptic connections.

The issue of indirect synaptic connections was addressed in the original discussion in the 'Identification of synaptically-connected PC-DCN cells pairs' subsection, however, the discussion of this issue has been expanded. In addition, a new section entitled, 'Technical Limitations' has been added to the Discussion to address the other issues raised by the reviewers and editor.

With regard to the number of DCN cells, it is true that the number of DCN cells that were analyzed in detail is small because only a few examples out of a large database could meet the criteria to be included, and we agree that this limits how much we can generalize from our results.

However, it is important to realize that the general relationship between the level of CS synchrony and the strength of inhibition of DCN was observed for a larger population of DCN cells as reported in our previous work (Blenkinsop and Lang, 2011) and that the smaller subset of DCN cells whose activity is analyzed here in greater detail rests on this initial finding. Moreover, we believe the relationships that were found for each DCN neuron whose activity was analyzed in the current paper are based on strong evidence from large amounts of data. Specifically, each such DCN cell, the analyses are based on thousands (~2500-12,000) of CSs in its presynaptic PCs and tens of thousands (~30,000-50,000) of its own spikes. Thus, we feel confident of the relationships that are described in the paper, and though we agree that the data are insufficient to say whether these are the only types of relationship that exist between DCN and PC CS activity, the fact that a significant synchrony-dependence was found for all four DCN cells that were analyzed suggests that synchrony is likely to be an important parameter in many cases.

Separate reviews (please respond to each point):

Reviewer #1:

This manuscript argues that the synchrony of complex spikes in the Purkinje neuron cohort projecting to the same DCN neuron might hold the key for controlling cerebellar output. The manuscript is written in clear style and the data are analyzed and presented well. My biggest worry is that the data presented are from only four or five DCN cells (Results paragraph one says five cells, but four experiments are presented. Authors need to clarify how data from two DCN cells in the same animal were combined). In their dataset, authors could identify 4-7 PCs to be synaptically connected to the DCN cell being recorded. This combined with the low number of highly synchronous CS inputs in each experiment (Figure 6B), places the findings on not-so-solid ground. In addition, of the four, one DCN cell (A) did not show the tight synchrony-dependent modulation seen in the other three cells (Figure 9B and 10). The authors speculate that this cell could be an IO projecting DCN but this idea is not tested. With this in mind, I am not certain that the main point the authors push, i.e. that CS synchrony in a 10ms window is the most important parameter for DCN output, can be accepted. Some specific points:

The reviewer raises a number of points related to the low number of DCN cells that were ultimately analyzed, including how data from two DCN cells recorded from the same animal were combined.

The issue of the number of DCN cells that could meet the criteria to be analyzed is addressed above.

The two DCN cells referred to by the reviewer were recorded separately. Thus, they were treated as separate cells and analyzed individually.

1) Figure 1 is not useful. Either remove or include PC recordings and a Venn diagram of cell numbers discussed at the end of paragraph one of the Results.

We feel it is generally important to show examples of the actual recordings to let the readers judge the quality of the raw data that was used for the subsequent analyses. Usually one or more 'representative' examples are given, however, given the limited number of DCN cells that are the focus of the paper, we thought it better to show all of them.

We have, however, modified the figure based on the suggestions of several of the reviewers. First, we now show each of the DCN cells using an expanded time scale so that the spike waveform of each can be seen. In addition, we now show only one example over a longer (5 s) period to demonstrate the typical firing pattern. Lastly, we have added an example of the CS activity recorded from one of the PCs.

In addition, the initial two paragraphs of the results, which describe the database, have been revised to make clear the relationship between the original dataset, which was first described in our previous paper (Blenkinsop and Lang, 2011), and the subset of cells that are analyzed in detail in the current paper.

2) Will be useful to indicate the long latency modulations of DCN activity in Figure 2, with an arrow or grey bar.

The histograms shown in Figure 2 are for individual cells, but the long-latency analysis was based on an analysis of the group activity. Thus, instead of showing the modulation periods on the histograms in Figure 2, a population histogram has been added to Figure 11 and the modulation periods are shown on it.

3) In Figure 3, authors show lines to indicate 95th and 99th percentile ML separation and state that the synaptic group value is above the 99th percentile. This should be the 5th and 1st percentile and "below the 1st percentile" given that the synaptic group is placed more closely than the others.

The changes were made to Figure 3.

4) Figure 4A: SD or SEM? Prefer that the authors would show the individual synchrony values of all PC's in that experiment as scatter with an indication of the mean. Similar plots can be made for all four experiments. Figure 4B does not make much sense to me. What does it mean that the ratio is several-fold over very long distances?

The error bars in Figure 4A indicate SEM. This is now stated in the figure legend.

We could not determine exactly what reviewer was referring to with the phrase 'the ratio is several-fold over very long distances', but Panel 4B shows the ratio of the two curves shown in 4A for each of the four experiments, and is simply a way to show how the presynaptic cells display higher levels of synchrony among themselves than do the remaining PCs at all separation distances. It is true that this information (the relative difference in synchrony) is implicit in 4A, but we think that showing it explicitly is also useful. In particular, doing this facilitates comparison of the relative synchrony (presynaptic vs other PCs) distributions across experiments. In addition, it more clearly shows that synchrony is better maintained with distance for the presynaptic PC group (indicated by the increase in the ratio with increasing separation).

This last result is important because it leads to another line of evidence supporting the correlogram-based identification of presynaptic PCs. This follows from two findings: that synchrony is better maintained within, as compared to between, zebrin bands (Sugihara et al., 2007) and that each DCN region receives PC input from only one zebrin band. Thus, the better maintained synchrony with distance suggests that the presynaptic PCs were more likely to located within the same zebrin band (otherwise their synchrony would drop off similarly to the other PCs, which cannot all lie within the same band), which fulfills the requirement of the second finding, which implies that PCs that project to the same DCN cell are from the same zebrin band. This argument has been added to the Discussion.

5) Figure 11B: Authors argue that the long latency modulation of DCN activity is also synchrony dependent but the effect is weak and variable. One of the four cells recorded did not show this modulation. This point appears weak and not related to the main point of the manuscript.

It is true that long latency modulation is not a major point of the paper, but this modulation was present in most cases, which raises the question of what its relationship to synchrony is, and so for completeness we believe it is worth including an analysis of it.

6) Results paragraph two, final sentence: What is the larger sample used in this comparison?

The larger sample refers to the DCN cells that were recorded in Blenkinsop and Lang, 2011. The citation to this work is listed following the firing rates that were given. We added text to the sentence to make this reference clearer.

7) Subsection “Inhibition of DCN activity varies with level of CS synchrony” paragraph five: insert "Figure 7F"

Figure reference was inserted.

8) Sixth paragraph of the same section: Could singular CS's still be high synchrony events synchronized with PCs not being recorded from?

Our presynaptic PC group is a sample of the population of PCs that project to the DCN cell. So, it is conceivable (and even likely) that some singular CS events in our sample population are part of high synchrony events in the whole population. However, the assumption is that the activity of the PC sample group represents that of the population (i.e., is a random sample), a standard, if not wholly testable, assumption of recording studies. To the extent that the presynaptic group is representative, the level of synchrony in the sample should reflect that of the population as a whole, on average.

9) Subsection “Synchrony may be needed to modulate DCN activity”: "Our results indicate that CSs need to be synchronized to be able to affect DCN activity". Given that singular CSs also show about 20% inhibition, one would argue that they are also able to affect DCN activity, albeit at a lower level relative to the synchronous ones. Again, as noted above, it is likely that the singular CSs are in fact a mixed population of events of widely varying synchronicity. This point needs to be brought out in the discussion.

The reviewer is correct that the singular CS events represent a mixed population of events (as mentioned in response to the previous point), and this may contribute to their showing an inhibitory effect. However, we do not believe this is a major factor for why a significant inhibition is observed with singular CS events.

The key point is that one the firing of PC in the group represents the firing of a significant fraction of our synaptic group firing (between 14% and 25% of the group depending on the experiment). In the Results section we show that if you extrapolate the relationship between synchrony and inhibition of the DCN cell back to a single PC firing a CS (to do this we assumed 50-100 PCs converge onto a single DCN cell), the inhibitory effect is negligible (i.e., not significantly different from zero).

Also, because our use of the word 'singular' (as a single member of the presynaptic group) could be misleading (i.e., taken to mean one PC out of the entire presynaptic population) we have replaced it with text that removes this ambiguity.

10) Subsection “CS synchrony: direct and indirect actions on the DCN” paragraph four: later

Fixed

Reviewer #2:

This manuscript is a meta-analysis of multielectrode data from recordings of Purkinje cells (PCs) and deep cerebellar nuclei (DCN) in anesthetized rats. The authors correlate activity of PC complex spikes and DCN spikes to find likely connected cells and then explore the effects of synchronized PC complex spikes on 4 DCN cells. The greater the degree of complex spike synchrony, the greater the inhibition of DCN cells. The manuscript has useful information and the authors are careful to report all the exceptions and special cases, but the text doesn't always make it clear what the relevant points are or what aspects of the data the authors wish to stress. There are relatively few points along the way through 11 figures where the results and interpretations are synthesized. It would be helpful if the text were re-written to include a few more summary sentences. Likewise, there are many figures with relatively few panels, but sometimes the transformations of the raw data aren't immediately obvious. Possibly some further illustration would be helpful. Specifics are given below.

Summary sentences were added to the Results subsections.

Comments.1) Figure 1 has example traces of 4 DCN cells, but nothing else, and it is hard to know what we are looking at them for. Are these the 4 cells that are identified in Figure 3 and carried through the paper? Figure 2 also shows 4 histograms from DCN cells but, in this case, 2 plots are from one DCN cell. PC spikes and PC synchrony are never actually illustrated. It could be helpful to see some of the PC traces (and maybe combine Figures 1 and 2 and relate them to 3) so that the raw data are clearer.

To clarify, the four DCN cells in Figure 1 are the ones analyzed throughout the paper. This is now stated. Examples of CS activity were not included originally because such examples were shown in several prior publications. However, we have now included an example.

The reason for showing each correlogram in Figure 2 with both a compressed (top row) and expanded (bottom row) x-axis is to show both the long-latency modulation (top row) and the precise timing of the CS-triggered inhibition (bottom row) that was used to define a synaptic relationship.

2) Figure 3 is used to make the argument that the likelihood of finding converging PCs decreases with mediolateral (M-L) but not rostrocaudal (R-C) distance. This argument is convincing in the M-L dimension but less so in the R-C dimension, since the multielectrode array has 7-9 electrodes M-L but only 3-4 R-C, and in Figure 3E and F the converging PCs have about equivalent mean M-L and R-C separations. Should this analysis be curtailed to talk about clustered PCs rather that PCs in an R-C plane?

We agree that the dimensions of crus2 and the recording array allow detection of synchrony changes over larger distances in the mediolateral direction than the RC direction. Nevertheless, we prefer to keep the separate analysis of the RC distance for several reasons. Despite the limited range in the RC direction, the resolution in both planes is the same (250 µm). The maximal separation in the RC direction that be detected was 750 µm, and so we should be able to detect a drop off in synchrony that occurs over shorter distances. Indeed, such a rapid drop off in synchrony occurs in the ML direction as shown in previous publications and in Figure 4, and so the lack of finding a difference in the RC direction is a useful contrast, and furthers the argument that convergent PCs form a RC oriented band.

3) The Synchrony Index described in the Materials and methods and used first in Figure 4 could be explained more clearly and intuitively, possibly even illustrated. The equation indicates that it is the summed product of time-matched deviations from the mean for each of two cells, divided by the root of the product of the summed-square-deviations. It is clear that when both signals go high simultaneously (i.e., synchronize), The Synchrony Index value will increase, but it is not immediately obvious what the range of values is expected to be, how this will vary with different levels of background firing, etc. It would be useful if the authors could help calibrate the reader, so that the values of Figure 4 (0.12, 0.06) can be placed in perspective. This figure might be a useful place to illustrate the successive transformations of the data.

We have added a brief description in the methods to provide the information the reviewer requested. The synchrony level, C(0), defined by the formulas in the Materials and methods is simply the cross-correlation coefficient between the two spike trains at 0 ms latency and this is stated in the Materials and methods. As such, it is bounded with a range of 1 to -1, where these limits reflect perfect correlation and inverse correlation, respectively, and a value of 0, represents no correlation. A detailed description of the statistical significance of particular values of C(0) has been presented previously, and this work is now summarized and cited. Essentially, values that would be expected by chance are one to two orders of magnitude smaller than those observed among the presynaptic PCs.

C(0) is normalized for firing rate. That is, more synchronous events will occur by chance when the neurons have higher firing rates, but the value of the terms reflecting each synchronous event are smaller (because the mean firing rates being subtracted are greater), which compensates for this effect. However, the issue of changing firing rates doesn't seem to be of major concern here as C(0) is not being compared across conditions (e.g., after giving a drug like harmaline or picrotoxin) in which CS rates have changed significantly.

4) Subsection “Inhibition of DCN activity varies with level of CS synchrony”; “Although singular CSs had the weakest inhibitory effect, they still reflect a significant level of synchronization….” It is not clear on what basis the authors say that the “singular” CSs, which seems to mean non-synchronized CSs, reflect synchrony. The argument about the 50-100 converging PCs is hard to follow, especially when the concluding sentence indicates that the effect is not different from no synchrony. Please clarify.

Since the term 'singular' could be misleading and taken to mean a complete absence of synchrony (i.e., the firing of only one PC in the entire population of PCs that project to a DCN cell) we have removed it. Instead, we now use a phrase like 'events in which only one PC in the group fires a CS'. This should make it clearer that we are referring to a single PC of the group of recorded PCs, which actually represents a significant fraction of the total population of PCs that are presynaptic to the DCN cell.

5) As far as this reader can tell, the central, most important finding of the manuscript is Figure 7F, which shows that for all four DCN cells, there is a strong correlation with steep slope of the amount of inhibition as a function of synchrony level. This key panel is not called in the text and hardly stands out in the illustrations. With so many figures and a relatively monotonic (without emphasis) presentation, it is possible that it will be missed. Some rewriting would help make this result more salient.

The figure reference has been added to the text. Also, the text in the section related to this finding has been revised.

6) The 7F panel is complemented by Figure 8. It switches to illustrating the inhibitory efficacy of PCs based on whether they are in a synchronous group, but the authors just refer to this as a “cell-by-cell” comparison, so it takes several readings before one realizes that this is a demonstration from the “perspective” of the PCs, which is quite important. Please rephrase for clarity. (Perhaps the authors could consider putting this key panel with Figure 7F somehow so that they stand out as a central figure?)

As the reviewer says, Figures 7 and 8 address distinct points, we believe they should remain as separate figures. However, we have revised the text to clarify the different points being made by Figures 7F and 8.

7) Discussion; “these [convergent] PCs have highly synchronized CS activity…” Is what is meant here that they have a high probability of synchronizing (which does not appear to have been tested; or was it?) or that when they synchronize they do so with <5 ms jitter (which is stated again in the last clause)?

What was meant is that they show more synchronous activity, which is demonstrated by the results related to Figure 4. The sentence was revised to eliminate the ambiguity related to the meaning of 'highly'.

Minor points.1) Results; “In a few cases (n=4)” Please indicate n=4 out of how many cases.

The number of cases (24) was added to the text.

2) Figure 2, please indicate bin width in legend.

The bin width (1 ms) was added to the legend.

3) Figure 4B, perhaps different line patterns (solid, dashed, dotted) could help distinguish the four curves.

We don't believe this is necessary, as the curves are distinct when they are separated from each other at the larger separations, and at the smaller separations the data points are virtually identical to each other, making it unnecessary to distinguish them.

4) The use of the word “singular” to describe non-synchronous CS's is rather non-standard and unintuitive. Perhaps a definition could be given at first use.

We eliminated the word and used a more precise phrase to define the event of one PC in the group firing a CS, as described in the responses to earlier comments.

5) Figure 5, it would be useful to indicate the number of PCs in each group in the legend or on the plots.

The numbers are now given in the legend.

6) Subsection “Inhibition of DCN activity varies with level of CS synchrony”; (Figure 7) “the baseline activity in all histograms was normalized for sweep number…” It is not clear what analysis was done. Please clarify.7) Please define “cusum.”

Cusum is now defined in the text.

8) The word “axonic” should probably be “axonal.”

Changed

Reviewer #3:

The authors of the manuscript entitled "Complex spike synchrony dependent modulation of rat deep cerebellar nuclear activity" investigated how Purkinje cell complex spike activity and synchrony affects activity in their main downstream target, the cerebellar nuclei (CN). By doing simultaneously recordings of Purkinje cells (PCs) in the cerebellar cortex and of CN neurons, they describe that PCs that are putatively presynaptic to recorded CN cells are located in rostrocaudal strips. In addition, they describe that CN cells show specific patterns of short-latency responses to complex spike activity of which the strength depended on the degree of complex spike synchrony. The paper addresses an important and timely topic using a series of detailed in depth analyses.I have several suggestions to improve the paper.1) The manuscript largely lacks raw data. Only in Figure 1, traces are shown but at a very compressed time scale. Please expand.

As requested, we have added records at an expanded time scale to show the individual DCN spike waveform.

2) The number of CN cells from which they recorded in a paired format with PCs is low (n = 4/5). Given that there are many different cell types in the nuclei (e.g., Dulac and Uusisaari studies), it may be worthwhile to increase the number of CN cells. In addition, the number of high-synchrony events per cell is low. This makes the conclusions, although likely to be accurate, somewhat preliminary.

We agree that the number of DCN cells is too low to allow strong statements about how the results may generalize to the various DCN cell types (in the discussion only a short speculation about DCN-IO neurons is made, and we clearly state that further evidence is needed to support this speculation). However, as discussed earlier, we believe that relationships that were found for each of the DCN cells that were analyzed rest on a strong foundation. Moreover, that all four cases showed a strong relationship between CS synchrony and the strength of DCN inhibition makes it unlikely that this is a rare phenomenon.

3) The authors discriminate between directly coupled Purkinje cell – DCN neuron pairs and non-synaptic pairs. Their criteria for establishing synaptic connections are not mentioned in the current paper, but a reference is made to a previous publication (Blenkinsop and Lang, 2011). Also in that publication, the criteria are not overly clear. In the latter study, the impact of indirectly coupled pairs has also been described (see Figure 4C of Blenkinsop and Lang, 2011). As the authors discuss (subsection “Identification of synaptically-connected PC-DCN cell pairs”), identification of synaptic connections is difficult and could be compromised by synchrony between Purkinje cells. As Purkinje cells with a high level of synchrony might be considered functionally equivalent, the authors rightfully remark that it is questionable whether this would affect their conclusions. However, the abovementioned, indirect connections (see Figure 4C of Blenkisop and Lang, 2011), could also affect the patterns. Thus, it is crucial to understand which criteria were used to define mono-synaptic connections and to what extent indirect connections could contribute to the findings.

The identification criteria of a monosynaptic connection from PC to DCN cell that we used are based on an accepted methodology (cross-correlogram analysis) for identifying such connections (e.g., Perkel et al., 1967; Aertsen et al., 1989; Csicvari et al., 1998). In our specific case there had to be a change in the correlogram at a latency that matched the conduction time of the pathway from the presynaptic (PC) to the postsynaptic cell (DCN). In the Materials and methods we state this criterion as, "a significant negative deflection in the CS-triggered correlograms within 5 ms after CS onset, consistent with the conduction time of the PC-DCN pathway and inhibitory nature of PCs". A more detailed discussion of the validity of this criteria and other evidence supporting identification of synaptic connections is now in the Discussion.

Regarding the issue of correlations among PCs, it is necessary to consider two distinct causes of these correlations: PCs contacted by branches of the same olivocerebellar axon and electrotonic coupling of olivary neurons. The first case would produce a near perfect correlation (r~1) with very little jitter in the timing of the spikes in the two PCs, and essentially would not affect the interpretation of the results for reasons elaborated in the discussion.

In contrast, CS synchrony due to the gap junction coupling between IO neurons, a 'high correlation' would typically result in correlations of <.2 and a central peak of the correlogram that spreads over 5-10 milliseconds. Thus, CS-triggered correlograms of DCN activity from PCs that do not synapse onto the DCN cell in question but whose CSs are correlated with those of a PC that does synapse onto the DCN cell, should show an onset of inhibition at negative latencies without any new effect at a latency matching the PC-DCN conduction time. Indeed, such correlograms are found (and were rejected).

4) Regarding the indirect connections: the authors focus on the synaptically connected pairs, but was there also an impact of complex spike firing in the non-synaptic group? According to Figure 4A (of the present manuscript), also these cells do show synchronous complex spike firing, albeit at a lower level, so that an effect would be predicted. Please discuss.

Testing whether the effect of 'indirect connections' is due to the correlation of the CSs of those PCs with the CSs of the presynaptic PCs is not easy to do, because the activity of both groups of PCs is correlated, as the reviewer notes. Indeed, we previously reported that the level of synchronization of PCs throughout the recording array could modify the inhibition of DCN activity related to the activity in presynaptic PCs. For the current paper, we attempted an analysis to test whether the effect seen with the cells not identified as presynaptic to the DCN cell was due to their activity being correlated with that of the presynaptic cells. Specifically, we treated the non-synaptic cells as a group and performed the same analysis as we did for the synaptic group. The first problem that arose is that this larger group has few or none of the higher synchrony events involving most or all PCs. This is not surprising, given the lower synchrony overall amongst these cells. In fact, it points to an immediate difference from the synaptic group. However, for the synchrony levels that did occur, one sees a good correlation between synchrony and the strength of inhibition of the DCN cell. Again, this is not surprising given that the non-synaptic PCs do show correlated activity with the synaptic ones. We then tried to decorrelate the activity of the synaptic and non-synaptic PCs by removing any events in which synaptic cells participated. When we did this the relationship between synchrony and DCN inhibition was weakened, often considerably. However, this procedure resulted in very few synchrony events being present at any level other than the lowest few levels, so that it was impossible to test the effect of this procedure at the high synchrony levels, which are the most critical to test.

5) The experiments were performed in anesthetized animals (with ketamine/xylazine). Since this anesthesia can affect cerebellar firing patterns including patterns of rhythmicity and synchrony (e.g., Schonewille et al., 2006), it will be worthwhile to replicate some recordings in awake behaving animals. This will increase the functional relevance of the study.

While we agree recordings from awake animals would be worthwhile, we believe obtaining them would be exceedingly hard technically, particularly given the difficulty in finding synaptically-connected pairs even in an anesthetized preparation. In the discussion we acknowledge this limitation but also provide reasons why it is plausible that the results should be relevant to understanding cerebellar function under more physiological conditions.

In particular, although some higher order statistics of SS, CS, and DCN activity may vary between ketamine/xylazine and awake states, on many measures their activity is similar. For example, their firing rates under ketamine/xylazine anesthesia are essentially the same as in the awake animal (note that the study of Schonewille, which only reports SS rates, specifically confirms this for SSs). Moreover, the phenomenon of long pauses of SSs under anesthesia that is a focus of the Schonewille report is not generally seen using our recording technique (see, Tang et al., 2017). Moreover, a more recent report from Schonewille and colleagues provided evidence that this activity pattern may actually be due to damage or cooling of the cortex (Zhou et al., 2015 J Neurophysiol 113: 2524-2536), and so raises the possibility that it is not directly relate to anesthesia. Finally, Schonewille, 2006, does not address differences in synchrony patterns, and CS synchrony patterns were shown to be essentially the same in ketamine/xylazine anesthetized and awake animals (Lang et al., 1999). In sum, the many similarities of PC and DCN activity in the anesthetized and the awake animal suggest that the present results will be relevant for physiological states.

6) In addition, or alternatively, the authors may want to discuss the Ten Brinke studies (2015 and 2017) in which complex spikes and CN activity were recorded during eyeblink conditioning. In these studies the same patterns were detected with separated single unit recordings of PCs and CN cells in awake behaving animals as described with simultaneous dual recordings in the current study. This would highlight the achievement of the current dual recordings of the current Lang study, while clarifying the putative relevance of it during natural behavior. In fact, the CS synchrony and impact thereof in the CN appear to be enhanced during motor learning.

The ten Brinke, 2017, paper is cited in the Discussion.

7) The authors relied on 1 ms binning of their data. The data underlying Figure 7 depend on the duration of the inhibition, which is defined as ending when the 3-point average reaches baseline levels again. This seems a very noise-prone analysis, especially for relatively rare occurrences of high-synchrony (see Figure 6). The open symbols in Figure 7B-E are therefore questionable; the closed ones a bit more reliable, but even here an estimate of the baseline firing during only 50 or 25 ms (the choice of either which value being not really justified), might be quite arbitrarily. Overall, the conclusion that more synchrony leads to more inhibition seems plausible, but the statistical evaluation could be improved (see also below). One should notice, again, that all these conclusions are based upon only four cells.

The periods defined by the various methods might vary somewhat because of noise; however, the overall results are not particularly sensitive to the precise limits, as shown by the overlap and similarity of the curves using different methods for defining these limits in Figure 7. We also tried using a fixed window of 5 to 60 ms for all experiments (so that the same time period could be compared across experiments), which again produced similar curves (data not shown). In sum, the relationships shown in Figure 7 did not greatly depend on the exact limits to do the analysis.

Additional data files and statistical comments:The authors report using paired and unpaired t-tests in addition to Pearson correlation analyses. However, their reasoning for choosing these tests is unclear. Based on the low n, inequality of sample sizes (e.g. the difference in number of low vs high synchrony events) and questionable normality of distributions, one may also have to consider non-parametric testing; this reviewer is not a statistician, but please ask advice from a professional to make sure the current approach is correct or not. For sure, the authors should include a detailed description of tests used (and for what reasons they were used) and also include the specific outcomes like t-values, degrees of freedom etc. Evaluation of the statistics may also indicate how many pairs of PC-CN cells need to be illustrated in the current paper to back up the conclusions. So this reviewer has no real doubts about the conclusions of the current manuscript, but they need to be properly backed with solid statistics to stand the test of time.

The reviewer is concerned about using t-tests because of low n, unequal sample sizes and questions about the normality of the distributions. The first two of reasons are by themselves not an issue, as differences in sample size and variance between populations can be taken into account in the formula used to compute the t-statistic and degrees of freedom for the t-test (sometimes referred to as the Welch test). This was done in all tests where differently sized populations were compared, and this is now stated in the Materials and methods.

Regarding the low n, a t-test was used in a handful of cases, and in only one case was the n particularly low (n=4) The n's of the other populations were: 24, 54, and 100. It is worth noting that t-tests were specifically designed for testing small samples, so a low n is not a problem in itself, but only may become an issue if the population deviates strongly from normality.

Nevertheless, to address the concern about normality, each of sample populations was tested for normality using the Kolmogorov-Smirnov (KS) goodness-of-fit test. If the populations were not normally distributed, a non-parametric test was used to compare the populations.

With regard to Pearson's correlation (r), Zar, 1999, states that the test for whether r is different from zero is robust with regard to violations of normality so that t-tests are appropriate and that transforms to normalize the data are necessary only if one wants to test whether r differs from some non-zero value (see section 19.2, Zar, 1999).

[Editors' note: further revisions were requested prior to acceptance, as described below.]

The manuscript is indeed written more accessibly than in the previous submission, but there are some remaining issues that we feel should be addressed before your paper is published, as outlined below:1) The authors claim (subsection “Technical limitations”) that DCN firing rates are the same with and without anesthesia. No citation is given. The 34 Hz DCN firing rate seen here seems relatively low. The main line of the authors' arguments doesn't rely on claiming that there is no effect of anesthesia, and so unless there is explicit evidence, it is probably better not to say that anesthesia does not affect the firing rates.

We agree that this point is not vital to the argument, but we nevertheless believe it supports the argument in that it provides some evidence that DCN cells are in a similar state in awake and anesthetized animals, and thus makes it more likely the operational principles of the system reported here are also valid for the system in the awake state.

To support our statement about DCN firing rates we have now given citations for the DCN firing rates in awake animals (see list below). Based on these references, 34Hz is well within the ranges of values reported for DCN neurons in awake animals, even if it is toward the lower end of the distribution. It is perhaps worth noting that DCN cells in mice seem to have very high rates, e.g., 85 Hz (Sarnaik and Raman, 2018); however, examples from other species often report much lower average values or a large range that extends well below 34 Hz.

The Discussion was modified to state the following:

“In particular, the reported spontaneous firing rates for DCN cells in alert animals varies widely, both within and between studies; overall, rates range from 10 Hz to greater than 100 Hz (e.g., Thach, 1968; Burton and Onoda, 1978; Harvey et al., 1979; Chapman et al., 1986; Fortier et al., 1989; Gruart et al., 2000; Sarnaik and Raman, 2018). The data set from our previous report showed a similar range for the overall DCN population as did the population of DCN cells for which a presynaptic PC was identified (see Figure 2A,B of Blenkinsop and Lang, 2011).”

Examples of spontaneous DCN firing rates reported for awake animals

monkeys

37 Hz, Thach 1968.

30-50 Hz, Harvey et al., 1979.

52 Hz, Chapman et al., 1986.

30-35 Hz Fortier et al., 1989

10-60 Hz and 30-80 Hz for cell types A and B, Gruart et al., 2000.

21 Hz, 16 Hz, and 29 Hz, depending on cell type, Burton and Onoda, 1978.

2) Figure 1: Reviewers wanted PC recordings to be shown here and this has not been done or addressed in the response.

Actually, this point was addressed in the previous response to reviewer 1 (the following was stated, "we have added an example of the [complex spike] CS activity recorded from one of the PCs") and panel E of the revised Figure 1 does show an example of CS activity from one of the PCs analyzed in the paper. Moreover, examples of CSs from the same parent data set can be seen in our previous publication (Figure 1 of Blenkinsop and Lang, 2011).

Perhaps the reviewers were equating PC activity with SS activity, but SSs were not recorded (see response to point 7 below).

3) Figure 4A: Individual PC synchrony values – not been done or addressed in the response (reviewer #1, point 4).

To include all of the individual synchrony values would make the plots in the figure difficult to read (i.e., in several cases 50-100 points of very similar values would be plotted for one x-axis value). So, we would prefer not to plot the data this way. However, to address the reviewer's concerns about the distribution of the values, we have described the statistics of the population in the text and added a new panel to the figure (new panel B), which plots the median synchrony values. See answer to point 6 below for details on the statistics used to describe the distribution and to test the significance of the differences between the synaptic and non-synaptic groups.

4) Figure 4: No illustration/schematic for data transformation used in this and subsequent figures, as suggested by Reviewer 2.

The synchrony index is simply the cross-correlation coefficient of two spike trains at zero time lag, calculated by standard formulas that have been published many times. The papers that first developed it and that first used it with regard to CS activity are cited for reference. We also added a description of its properties in the Materials and methods (this follows the equations). Given this information, we do not see what would be added by showing a schematic of the calculations.

5) No response provided for Point #6 from reviewer 2.

We have modified the sentence to read, "To test this possibility, the baseline activity in all histograms was normalized for (divided by) the number of CSs used to generate them and then compared (Figure 7G). "

6) Figure 4: No statistical tests seem to have been done?

In the previous version of Figure 4A, the results from a single experiment were shown. Although statistical tests could be done to test the significance of the differences in the example that was shown, in other cases the number of cell pairs in a particular group from an individual experiment was too small to perform a statistical test. So, we adopted a different approach that would allow testing of all the data, which is shown in the new panels A and B of Figure 4. We combined the cell pairs from all four experiments, which then allowed statistical comparisons for the population at separation distances of 0, 250, and 500 µm, which are the distances for the vast majority of presynaptic cell pairs (~95%, 60/63 pairs). In the revised figure, mean synchrony as a function of ML separation is compared in 4A for the populations of presynaptic and non-presynaptic cell pairs. t-tests showed that the means were significantly different at these three distances. The p-values and n's are given in the text. Tests for normality showed that the synchrony values were normally distributed at all three distances for the presynaptic population, but not for non-synaptic population. Thus, we added panel B, which shows the median synchrony values corresponding to the means shown in 4A and performed nonparametric tests (Wilcoxon-Mann-Whitney) on the distributions. These tests also indicated that the presynaptic and nonsynaptic distributions were significantly different at each distance. The p-values and n's are given in the text.

7) Figure 5: Will be better to plot the different lines in different colors, especially for Figure 5E where the spread around the mean can also be shown.

The curves in Figure 5E are now plotted in different colors and error bars indicating the SD are now plotted.

Results paragraph three: Were SS firing rates compared?

No, SSs were not recorded in this study. To make clear why only CSs were recorded, we now state the following in the Materials and methods section.

“Electrodes were implanted into the molecular layer to a depth of 100-150 µm below the folial surface, where CSs but not SSs can be observed. To record SSs, the electrodes need to be implanted to a depth of ~225-250 µm, close to the PC layer in rats. While also obtaining SS recordings is desirable, there are several reasons why this was not done. These include a much longer implantation process, greater overall damage to the cortex, more difficult spike sorting for analyses, and less stability during recording (i.e., the electrode is much more likely to damage or kill the cell being recorded because of its closer proximity to the cell body).”

Subsection “Inhibition of DCN activity varies with level of CS synchrony” paragraph nine and elsewhere: Wherever p values are given, please state the n's and the test used.

The n's are now stated with the p values. The default type of test used for each statistical comparison is stated in the Materials and methods rather than repeated for each instance. Where ever a non-default test is used, the specific test is named with the p-value.

Subsection “What is the effect of non-synchronized CS activity?” first paragraph: "This is not…" Odd sentence construction

The sentence has been modified to read as follows:

“This is simply because one PC represented a significant fraction of the presynaptic group (0.14 – 0.25) and its firing a CS thus likely corresponds, on average, to the synchronous firing of a similarly significant fraction of all of the PCs that project to a DCN cell fire CSs.”

Subsection “Identification of synaptically-connected PC-DCN cell pairs” and response letter: Authors argue that earlier work (Sugihara, 2007) shows that CS synchrony is higher within a zebrin compartment than between. However, that paper showed that this was true only for central and lateral crus2a. In medial crus2a, wide synchrony bands covering more than one zebrin compartment were observed. In the present study, it is not clear from which region of crus 2a neurons were recorded from.

Regardless of the exaction locations that PCs were recorded from (but see below for likely locations) we believe that the finding of high synchrony does provide supporting evidence for the cross-correlation based identification of PC-DCN connections. However, it is not absolute proof of their validity, and wasn't claimed to be, because of the possibility of false positives.

The reviewer is correct in stating that in the Sugihara et al. (2007) paper compartmental restriction of high synchrony was demonstrated for much of crus 2, but not for its most medial aspect. However, the implication that compartmental restriction was shown not to occur on medial crus2 is not correct. It is more accurate to say that paper showed it to be true for central and lateral crus 2a but was unable to determine whether it was also true for the medial crus 2a because of technical limitations (i.e., the medial compartments are too narrow to test adequately). Indeed, in the example that was shown (Figure 3 of that paper) one can see that no electrodes were definitively implanted to zebrin positive bands that neighbor the negative bands in the medial region. Also, note that two of the three examples in that figure do show that the highest synchrony levels are found among PCs within the same band, but because of the technical issues, the conclusions were conservative.

The key point is that PCs within the same compartment showed high synchrony where it was adequately tested. Of course, this pattern may not hold elsewhere, but given the similarity of cerebellar and olivocerebellar organization throughout, the most plausible assumption is that it does hold. This, when combined with the fact that each region of the DCN receives virtually all of its input from either a zebrin positive or negative compartment, implies that PCs that project to the same DCN cell should show high synchrony. Thus, testing for synchrony levels becomes a reasonable test for the validity of the cross-correlation based identification of the presynaptic PC group, and had the presynaptic PCs not shown high synchrony it would have questioned the validity of their identification. Conversely, since they did show high synchrony it supports it (i.e., is consistent with their identification as all connecting to the same DCN cell).

High synchrony could represent a false positive if there were cases in which a group of PCs show a pattern of high synchrony unrelated to the zebrin bands. This may be what the reviewer is concerned about, as there are generally high synchrony levels on medial crus 2a. However, as just discussed, this doesn't preclude a banding pattern being superimposed on the baseline synchrony level.

Regardless, although the exact location of the recording array on crus2 was not determined in the present study, it is that the array was mainly or entirely over the central and lateral portions of crus 2. Moreover, it is even more likely that the synaptic PCs were located in these portions of crus 2a (zebrin bands 4a- through 5a-, as defined in (Sugihara et al., 2007b). Medial crus 2a projects to various parts of the medial DCN. In contrast, neurons with histologically identified locations in the original data set were almost exclusively (one exception was on the lateral border of the medial DCN) located in the interposed and lateral DCN (see Figure 9 of Blenkinsop and Lang, 2011). Furthermore, DCN cells (including two of the cells analyzed here) that had synaptic connections with PCs were all in the middle to lateral interposed nuclei, which indicates that the synaptic PCs would be in the central and lateral crus 2a (zebrin bands 5+ through 7+, as defined in (Sugihara et al., 2007b).

Subsection “Identification of synaptically-connected PC-DCN cell pairs” paragraph five and elsewhere in discussion: "…as it is well-known that there is a pause in SSs following a CS," – A CS may have diverse effects on SS firing such as facilitation, pausing or suppression (De Zeeuw et al., 2011) or state switches (Loewenstein et al., 2005; Sengupta and Thirumalai, 2015). In this and other places in the manuscript, the writing seems to suggest that CS leads only to SS pauses.

We disagree with this characterization of what was written. For example, in the Discussion we state, "The latter includes the pause in SSs following a CS and the further modulation of SS activity following the pause" and "following its post-CS pause, SS activity often rebounds to rates that are significantly higher than baseline for 50 – 100 ms". Thus, we clearly discuss other aspects of how CSs affect SS activity. It is worth noting that the SS modulations mentioned by the reviewer always follow the initial pause in SS activity that follows a CS. In fact, if to definitively assign spikes as PC SSs you need to see this pause, otherwise it is possible that they are spikes from another cell type (e.g., basket or stellate).

In the specific line referenced by the reviewer, possible causes of the changes in activity of the DCN cells during the time immediately following the CS are being discussed. The post CS SS pause is the focus there, because it occurs during that time period. The other types of modulation that the reviewer raises largely occur after this time period, and so are not relevant to that particular discussion. They are discussed at other points in the discussion, as shown by the example quotations above.